


# Long-term trends of surface ozone and its influencing factors at the Mt. Waliguan GAW station, China, Part 2: Variation mechanism and links to some climate indices

Wanyun Xu[1], Xiaobin Xu[1], Meiyun Lin[2], Weili Lin[3], Jie Tang[3], David Tarasick[4], Jianzhong Ma[1], Xiangdong Zheng[1]

[1]State Key Laboratory of Severe Weather &Key Laboratory for Atmospheric Chemistry, Institute of Atmospheric Composition, Chinese Academy of Meteorological Sciences, Beijing, 100081, China
[2]NOAA Geophysical Fluid Dynamics Laboratory and Program in Atmospheric and Oceanic Sciences in Princeton University, Princeton, New Jersey 08540, USA
[3]Meteorological Observation Center, China Meteorological Administration, Beijing, 100081, China
[4]Science and Technology Branch, Environment Canada, 4905 Dufferin Street, Downsview, Ontario, M3H 5T3, Canada

*Correspondence to*: Xiaobin Xu (xuxb@camscma.cn)

**Abstract**

Interannual variability and long-term trends of tropospheric ozone are both of environmental and climate concerns. Ozone measured at Mt. Waliguan Observatory (WLG, 3816 m asl) on the Tibetan Plateau over the period 1994-2013 has increased significantly by 0.2-0.3 ppbv year$^{-1}$ during spring and autumn, but shows a much smaller trend in winter and no significant trend in summer. Here we explore the factors driving the observed ozone changes at WLG using backward trajectory analysis, chemistry-climate model hindcast simulations (GFDL-AM3), a trajectory-mapped ozonesonde dataset and various climate indices. A stratospheric ozone tracer implemented in GFDL-AM3 indicates that stratosphere-to-troposphere transport (STT) can explain ~70% of the observed springtime ozone increase at WLG, consistent with an increase in the NW air mass frequency inferred from the trajectory analysis. Enhanced STT associated with the strengthening of the mid-latitude jet stream contributes to the observed high-ozone anomalies at WLG during the springs of 1999 and 2012. During autumn, observations at WLG are more heavily influenced by polluted air masses originated from Southeast Asia than in the other seasons. Rising Asian anthropogenic emissions of ozone precursors is the key driver of increasing autumnal ozone observed at WLG, as supported by the GFDL-AM3 model with time-varying emissions, which captures the observed ozone increase (0.26±0.11ppbv year$^{-1}$). AM3 simulates a greater ozone increase of 0.38±0.11 ppbv year$^{-1}$ at WLG in autumn under conditions with strong transport from Southeast Asia and shows no significant ozone trend in autumn when anthropogenic emissions are held constant in time. During summer, WLG is mostly influenced by easterly air masses but these trajectories do not extend to the polluted regions of eastern China and have decreased significantly over the last two decades, which likely explains why summertime ozone measured at WLG shows no significant trend despite ozone increases in Eastern China. Analysis of the Trajectory-mapped Ozonesonde dataset for the Stratosphere and Troposphere (TOST) and trajectory residence time reveals increases in direct ozone transport from the eastern sector during autumn, which adds to the autumnal ozone increase. We further examine the links of ozone variability at WLG to the QBO, the North Atlantic Oscillation (NAO), the East Asian summer monsoon (EASM) and the sunspot cycle. Our results suggest that the 2-3 year, 3-7 year and 11 year periodicities are linked to QBO, EASMI and





NAO and the sunspot cycle, respectively. A multivariate regression analysis is performed to quantify the relative contributions of various factors to surface ozone concentrations at WLG. Through an observational and modelling analysis, this study demonstrates the complex relationships between surface ozone at remote locations and its dynamical and chemical influencing factors.

## 1    Introduction

Ozone in the troposphere is a potent greenhouse gas, an air pollutant detrimental to human health and vegetation, and the primary source of hydroxyl radicals, which play a critical role in atmospheric chemistry. The long-term variation of ozone is both of environmental and climate concern. Therefore, it is important to trace the long-term variations of ozone at different

locations and understand the causes of such variations. Continuous long-term observations of surface ozone have been made only at a few representative sites in China. The Mt. Waliguan (WLG) station, established in 1994, is situated in the northeastern part of the Tibetan Plateau, where population is scarce and industries hardly exist. It is one of the baseline stations in the Global Atmospheric Watch (GAW) program of the World Meteorological Organization (WMO) and the only one in the hinterland of the Eurasian continent with surface ozone measurement of longer than two decades. The long-term trend and periodicity of

surface ozone at WLG during 1994-2013 are reported in the companion paper (Xu et al., 2016). Significant increasing trends have been detected in spring and autumn, while observations show no significant trend during summer when ozone is at its seasonal maximum. Here we investigate the mechanisms controlling the seasonal ozone trends measured at WLG using backward trajectory analysis and multi-decadal hindcast simulations (1980-2014) conducted with the Geophysical Fluid Dynamic Laboratory chemistry-climate model (Lin et al., 2014; 2015a; 2017).

With the rapid economic development in Eastern China, the anthropogenic emissions of ozone precursors have been increasing during the past two decades (van der A et al., 2006;Kurokawa et al., 2013 ;Itahashi et al., 2014). Specifically, emissions of $NO_x$ over Eastern China have tripled since 1990 (e.g. Lin et al., 2017). Increasing levels of surface and free tropospheric ozone have been detected at several locations in eastern China, e.g. in the North China Plain (Ding et al., 2008;Wang et al., 2012;Ma et al., 2016;Sun et al., 2016;Wang et al., 2017a), in the Yangtze River Delta (Xu et al., 2008) and in the Pearl River Delta

(Wang et al., 2009b;Lin et al., 2017). Transboundary pollution transport from continental Asia has been implicated in raising Surface ozone level in the western U.S. is also often affected by transpacific transport of ozone and its precursors from East Asia (Verstraeten et al., 2015;Lin et al., 2015b;Lin et al., 2017). A recent study by Lin et al. (2017) shows that a tripling of $NO_X$ emissions in Asia contributed up to 50-65% of the observed springtime ozone increases at western U.S. rural sites during 1988-2014, outpacing ozone decreases resulting from U.S. domestic $NO_X$ emission controls. A few case studies documented

the influence of anthropogenic pollution from eastern and central China on ozone at WLG during the summer season (Wang et al., 2006b;Xue et al., 2011). Whether the growth in East Asian anthropogenic ozone precursor emissions has contributed to the long-term trend of surface ozone at WLG needs to be examined.



Asides from regional precursor emissions and long-range horizontal transport (Wang et al., 2006a;Lal et al., 2014), the concentration of surface ozone has many other influencing factors. For instance, surface ozone concentrations at high-elevation sites can also be raised by the downward transport of ozone-rich air from the stratosphere during deep convection and stratosphere-to-troposphere exchange (STE) events (Bonasoni et al., 2000;Stohl et al., 2000;Lefohn et al., 2012;Jia et al.,

2015;Ma et al., 2014;Langford et al., 2009;Langford et al., 2015;Lin et al., 2012a;Lin et al., 2015a). Studies based on short-term measurements suggested that surface ozone at WLG  is influenced by STE events in spring (Zheng et al., 2011) and sometimes during summer (Ding and Wang, 2006). The extent to which STE influences observed year-to-year variability and decadal trend of ozone at WLG has not been investigated.

Changes in large-scale atmospheric circulation patterns can modulate long-range transport of ozone pollution in the

troposphere as well as stratosphere-to-troposphere ozone exchange. The large-scale physical and dynamical processes including the El Niño–Southern Oscillation (ENSO), the North Atlantic Oscillation (NAO), the Quasi Bi-annual Oscillation (QBO) and the solar cycle (Creilson et al., 2003;Ziemke et al., 2005;Oman et al., 2013;Sioris et al., 2014) have been found to significantly affect stratospheric and tropospheric ozone variability. Based on the good correlation between the ENSO index and tropospheric column ozone (TCO) over tropical latitudes, Ziemke et al. (2010)created a so-called "Ozone ENSO Index"

(OEI). Over northern mid-latitudes, strong El Niño events enhance long-range transport of Asian ozone pollution towards the eastern North Pacific and the southwestern U.S. by modulating the strength and position of the subtropical jet stream (Lin et al., 2014). Some studies (Zeng and Pyle, 2005;Langford, 1999;Koumoutsaris et al., 2008;Voulgarakis et al., 2011) suggested that the change in dynamics after El Niño events can promote the cross-tropopause ozone exchange and lead to a rise in global mean tropospheric ozone burden. However, Lin et al. (2015a) found that El Niño events lead to enhancements in upper

tropospheric ozone but this influence does not reach surface air. Over western U.S. high elevation regions prone to deep stratospheric intrusions, Lin et al. (2015b) found that the increased frequency of deep tropopause folds that form in upper-level frontal zones following strong La Niña winters exerts a stronger influence on springtime ozone levels at the surface than the El Niño-related increase in lower stratospheric to upper tropospheric ozone burden.

Similar to the western US, the Tibetan Plateau has been identified as a preferred region for deep stratospheric intrusions

(Škerlak et al., 2014). Prior studies show that the QBO, ENSO and sunspot cycle influence the total ozone over the Tibetan Plateau (Ji et al., 2001;Huang et al., 2009;Ningombam, 2011;Zou et al., 2001). The mechanisms controlling interannual variability of jet stream characteristics, STE and their influences on lower tropospheric ozone measured at WLG are poorly characterized. In addition, China is largely influenced by the East Asian summer monsoon (EASM). Past studies point out that the EASM influences ozone concentrations in this region through altering the transport of anthropogenic pollution (Derong et

al., 2013;Liu et al., 2009;Liu et al., 2011). Yang et al. (2014) used the GOES-CHEM global chemical transport model to examine the impact of the EASM on surface ozone all over China and showed that a significant positive correlation exists between the EASM and the Tibetan surface ozone concentration, which can lead to inter-annual variations of 5-10%. However, their results were not evaluated with actual observations of surface ozone.



In this work, we aim to advance knowledge on the factors driving interannual variability and long-term trends of ozone at WLG over the past three decades. Sect. 2 briefly describe observational data, model simulations and analysis approach. In Sect. 3, we first discuss the links of surface ozone at WLG to air mass origin, including their seasonal to interannual variability (Sect. 3.1). We then use the GFDL-AM3 model hindcast simulations to interpret the influences of changes in meteorology,

STE and anthropogenic emissions on ozone measured at WLG in winter, spring, summer and autumn (Sect. 3.2). The impact of direct ozone transport versus precursor transport is also discussed (Sect. 3.3). Sect. 4 examines the relationship between atmospheric dynamics and surface ozone at WLG, including the influence of STE, NAO, EASM and sunspot cycle. An empirical model is obtained for normalized monthly level of surface ozone at WLG using the multivariate regression technique and used to explain the observed ozone trends.

**2   Data and Methodology**

**2.1  Data**

Ozone concentration, UV and meteorological parameters were measured at the Mt. Waliguan site (WLG, 36º17' N, 100º54' E, 3816 m asl) in Qinghai Province, China. Ozone concentrations at a5-min resolution from Aug. 1994 to Dec. 2013 were averaged into hourly, daily and monthly resolutions, with a data completeness of 75% required for each averaging step. The

Hilbert-Huang Transform (HHT) analysis was performed respectively using the monthly average daytime and nighttime ozone data. Further details on the site, measurements, the daytime and nighttime data subsets and the HHT calculations can be found in Part I of our study (Xu et al., 2016). In this paper, the results of the HHT analysis are further associated with the sunspot number (SSN), the QBO, the NAO and the EASM index (Li and Zeng, 2002) to investigate the influence of various climate oscillations on surface ozone at WLG. The SSN data are from Sunspot Index and Long-term Solar Observations

(http://www.sidc.be/silso/datafiles). The QBO index is the 50 mb Singapore zonal wind and available at http://www.cpc.ncep.noaa.gov/data/indices/qbo.u50.index. The NAO by (Jones et al., 1997) were obtained from https://www.esrl.noaa.gov/psd/data/correlation/jonesnao.data. The EASM index can be acquired from http://ljp.gcess.cn/dct/page/65577.

**2.2  Backward Trajectory Analysis**

The HYSPLIT model (version 4) from NOAA Air Resources Laboratory (Draxler and Hess, 1997;Draxler and Hess, 1998;Draxler, 1999) is used for the trajectory analyses, using three different meteorological datasets from NCEP. The NCEP Global Reanalysis Data with a spatial resolution of2.5°,the NCEP FNL operational data in 1.0° resolution and the NCEP GDAS (Global Data Assimilation System) operational forecast data in 1.0° resolution are used for 1994-1996,1997-2006 and 2007-2013, respectively. All the reanalysis meteorology data have a temporal resolution of 6-h. The trajectory endpoint is set





to 36.28º N and 100.90º E with a height of 100 m above the ground level. The 168-h (7 days) backward trajectories are calculated at a 6-h interval from 1st Aug. 1994 to 31st Dec. 2013.

To study the overall air-mass origin and to determine whether the air-mass collected pollutants from the nearby cities, the direction of each trajectory start-point relative to the WLG station is calculated both for the 168-h and for the 24-h trajectory.

The directions of start-points relative to WLG are clustered into bins of 45° and the occurrence frequency in each bin is calculated.

Since ozone is a trace gas with a distinct vertical distribution, it is not enough to just determining the direction of where the air-mass came from. The height of the air-mass is also very crucial for interpreting the measured ozone concentrations. The planetary boundary layer (PBL) height along the trajectories is used to judge whether the air-mass that arrived at WLG is

representing the PBL or the free troposphere. PBL trajectory sections are defined as the part of the trajectory that was continuously within the PBL before arriving at the station. Thus, PBL trajectory sections are usually close to the station. When the trajectory height exceeds that of the PBL, the rest of the trajectory is taken as the free troposphere trajectory section. Free tropospheric trajectory sections can also be close to the station, representing subsiding air from the free troposphere near the station, however, most of them cover the sections that are located far away from the station.

**2.3  Potential source contribution function analysis**

The potential sources of high ozone are studied using the potential source contribution function (PSCF) analysis method, which has been widely applied to detect possible source regions (Ara Begum et al., 2005;Lucey et al., 2001;Zhou et al., 2004). The PSCF analysis is performed both on PBL and on free tropospheric trajectories to detect differences in source region distributions in the PBL and in the free troposphere or above.

The PSCF on the grid (i,j) is defined as:

$$PSCF = m(i,j)/n(i,j), \tag{1}$$

where $n(i,j)$ is the residence time of all the trajectories and $m(i,j)$ is the residence time of a subset of trajectories in the grid$(i,j)$. Each trajectory is associated with ozone concentrations that were measured at its arrival time. The 75th percentile of all the ozone concentrations at WLG is calculated and the residence time in each grid $m(i,j)$ of the subset of trajectories that is

associated with ozone concentration higher than the 75th percentile value is counted.

Abnormally high PSCF values may be produced for certain grids with small $n(i,j)$ values, which would bear large uncertainties. To avoid such uncertainties, a weighting factor $W(n_{ij})$ is introduced, which was originally proposed by Zeng and Hopke (1989):

$$W(n_{ij}) = \begin{cases} 1.0, & n_{ij} > \overline{n_{ij}} \\ 0.7, & 0.1 \cdot \overline{n_{ij}} < n_{ij} \leq \overline{n_{ij}} \\ 0.4, & 0.05 \cdot \overline{n_{ij}} < n_{ij} \leq 0.1 \cdot \overline{n_{ij}} \\ 0.2, & n_{ij} \geq 0.05 \cdot \overline{n_{ij}} \end{cases}, \tag{2}$$

where $\overline{n_{ij}}$ is the average number of $n_{ij}$.



### 2.4 Calculation of direct ozone transport contribution

A Trajectory-mapped Ozonesonde dataset for the Stratosphere and Troposphere (TOST) was generated from the ozone sounding records by trajectory mapping byLiu et al. (2013).The dataset has a spatial resolution of 5º×5º×1km (latitude, longitude and altitude). A subset from the TOST, the global three-dimensional (3-D) monthly average tropospheric ozone from

1994 to 2012 is applied in this paper to calculate the contribution of direct tropospheric ozone transport to the ozone trends at WLG. The 3-Dtropospheric ozone data are in monthly intervals. A 3-D backward trajectory residence time within the same grids and with the same time interval as that of the tropospheric ozone data is calculated based on the backward trajectory analysis results in sect. 2.3. Assuming there is no ozone loss on the transport pathway, the contribution of $O_3$ in each grid to that of WLG during each month is calculated using the trajectory residence time as a weighting factor:

$$O_{3,contrib}(t,i) = \frac{O_3(t,i) \times T(t,i)}{\sum_{i=1}^{n} T(t,i)},$$    (3)

where $O_3(t,i)$ and $T(t,i)$ respectively stand for the ozone concentration and the trajectory residence time at time $t$ in grid $i$, while $n$ stands for the total number of grids.

To obtain the monthly time-series of total direct tropospheric ozone transport contribution to ozone at WLG, the 3-D ozone contribution climatology of all the grids is summed up (eq. 4). The bottom layer of the grid in which WLG resides is excluded

from the summation.

$$O_{3,contrib,tot} = \sum_{i=1}^{n} O_{3,contrib}(t,i)$$    (4)

The variation trend of $O_{3,contrib}(t,i)$ and $O_{3,contrib,tot}$ is calculated for the entire period of 1994-2013 and separately for each season. For display, $O_{3,contrib}(t,i)$ and its trend is integrated over height. $O_{3,contrib,tot}$ is used in the multivariate regression as $O_{3,trop}$ in Sect 2.6.

### 2.5 Modelling of stratospheric and anthropogenic contributions

The GFDL-AM3 global chemistry–climate model was used to make hindcast simulations of ozone and related tracers at ~200x200 km$^2$ horizontal resolution over the 1980-2014 period (Lin et al., 2015a;Lin et al., 2015b;Lin et al., 2014;Lin et al., 2017). The model is nudged to the NCEP/NCAR reanalysis zonal and meridional winds using a pressure-dependent nudging technique (Lin et al., 2012b). Two AM3 simulations are used in this study: one with both meteorology and anthropogenic

emissions varying from 1980 to 2014 (BASE) and the other with anthropogenic emissions (including methane) held constant in time (FIXEMIS). To quantify the stratospheric influence on surface ozone, a stratospheric ozone tracer ($O_3$Strat) is defined relative to a dynamically varying tropopause and is subjected to chemical and depositional loss in the same manner as odd oxygen of tropospheric origin (Lin et al., 2015a). Carbon-monoxide-like tracers for East Asia (EACOt), Europe (EUCOt) and North America (NACOt) are implemented to investigate the impact of circulation changes on hemispheric pollution transport

(Lin et al., 2014).  These CO-like tracers have a 50-day exponential lifetime and are simulated with surface emissions held constant in time from each of the three northern mid-latitude source regions. Comparison with available observations from the




mid-1990s to the 2000s at a suite of sites across Asia shows that GFDL-AM3 captures 65-90% of the observed ozone increases in Asia (Lin et al., 2017). The long-term ozone observational record at WLG provides an important test for the GFDL-AM3 model to represent the key processes driving year-to-year variability and trends of tropospheric ozone in the remote atmosphere of the Tibetan Plateau. For comparison with measurements at the 3.8 km altitude of WLG, the model is sampled at the 700hPa layer.

### 2.6 Multivariate regression of surface ozone at Waliguan

Multivariate regression is applied to obtain an empirical model to explain the relationship and contribution of the various influencing factors to the surface ozone concentration at WLG. The regression model takes on the following form:

$$O_3 = \alpha(t) + \sum_{i=1}^{n} \beta_i(t) \cdot factor_i(t) \tag{5}$$

Where $\alpha(t)$ is a third order harmonic function used to interpret the background variation signal of surface ozone:

$$\alpha(t) = a_0 + \sum_{j=1}^{3} a_{1,j} \cdot cos\big(2\pi j(t - t_0)\big) + a_{2,j} \cdot sin\big(2\pi j(t - t_0)\big) \tag{6}$$

and $\sum_{i=1}^{n} \beta_i(t) \cdot factor_i(t)$ stands for the total contribution of the $n$ influencing factors used in the regression model. $\beta_i(t)$ is a first order harmonic function, which can be expressed as:

$$\beta_i(t) = b_{i,0} + b_{i,1} \cdot cos\big(2\pi(t - t_0)\big) + b_{i,2} \cdot sin\big(2\pi(t - t_0)\big) \tag{7}$$

The coefficients $b_{i,1}$ and $b_{i,2}$ allow for the time-dependent intensification or attenuation of the influences of factors. Since our data start in the year of 1994, $t$ is calculated as:

$$t = year + (month - 1)/12 - 1994 \tag{8}$$

## 3 Key drivers of long-term ozone trends at WLG

### 3.1 Climatology and interannual variability of air mass origin at WLG

Based on past studies, which were mostly focused on summertime ozone at WLG, high ozone concentrations were mostly linked to downward transport, instead of horizontal transport of anthropogenic pollution (Zhu et al., 2004;Ma et al., 2005;Wang et al., 2006b;Xue et al., 2011). Westerly trajectories were commonly associated with downward transport events and high ozone concentrations, whereas easterly trajectories carried air masses with signals of anthropogenic pollution and lower ozone concentrations. Anthropogenic impact was attributed mostly to the two big cities, Xining and Lanzhou, which are located both to the east of WLG, however, central and eastern China could also have potential impacts (Wang et al., 2006b;Xue et al., 2011). Since ozone and its precursors are usually inhomogeneously distributed, both the horizontal direction and vertical height of the air-mass origin may impact the local concentration of ozone at WLG. To locate the origin of high surface ozone concentrations at WLG, a PSCF analysis (section 2.3) was performed separately for the PBL and FT trajectories. Figure 1 displays the ozone PSCF of the PBL and FT trajectories in spring, summer, autumn and winter from Aug. 1994 to Dec. 2013.





For the PBL as well as the FT trajectories, the NW sector is most frequently accompanied by high ozone concentrations (higher than 75$^{th}$ percentile ozone concentration), which is a common phenomenon existing in all seasons.

During spring, Sichuan Province, which is southeast to WLG, displays significantly high ozone PSCF both in the PBL and FT trajectories, which is possibly an evidence for long range transport of ozone and/or its precursors from Sichuan to WLG.

During summer, when air-masses from the east occur most frequently, the entire eastern sector reveals low PSCF, hardly showing signs of anthropogenic influence on WLG. High ozone PSCF occurs dominantly with trajectories from the NW or N. In autumn, significant contributions of trajectories from the E, SE and S can be discerned in the PBL trajectories, which suggest that high ozone is linked to air-masses coming from central China and the southeastern part of the Tibetan Plateau, as well as the southwestern part of Gansu province. In the FT trajectories, high ozone concentrations were also linked to air-masses from

central China. In addition, Gansu province and part of the Sichuan Province also shows high PSCF. In winter, the PBL trajectories show high ozone PSCF mainly in the NW directions, however, the SW and N-NE sectors also revealed scattered high PSCF values. Aside from the NW sector, the FT trajectories display significantly high PSCF in the NE sector in the western half of Inner Mongolia.

To evaluate the impact of different air-masses, we need to find out which air-masses are influencing WLG and evaluate the

relative importance of the different air-masses during different seasons. Figure 2 depicts the trajectory direction occurrence frequencies at t=-24h and t=-168h for spring, summer, autumn and winter, respectively. The t=-168h trajectory direction provides us information on the overall origin of the air-mass, while the trajectory direction calculated for t=-24h should be able to reveal if the air-mass passed over nearby polluted regions before arriving at the station.

From the t=-168h trajectory direction occurrence frequency (Figure 2 a2-d2) it can be seen that WLG is under the major

influence of western and north-western air-masses throughout the year, with air-masses from the east only playing a significant role during summer, which is in accordance with previous studies (e.g. Zhang et al. (2011)). Hence, the WLG site is overall very clean and highly representative of a background state. Trajectories from the east (including NE, E and SE) take up on average 15%, 57%, 26% and 4% of all the trajectory directions during spring, summer, autumn and winter, respectively. The NW trajectories at t=-168h are most frequent in spring (49%) and least frequent in winter (15%), while western trajectories

are most dominant in winter (70%) and least so in summer (6%).

From the t=-168h trajectory direction frequencies, it can be seen that the anthropogenic influence is negligible in all seasons except summer. However, the t=-24h trajectory direction frequency shows a significantly larger portion of easterly trajectories and smaller percentage of northerly trajectories (Figure 2 a2-d2), indicating that air-masses originating from the north of WLG often bend over to the east 24h before arriving at WLG. Trajectories from the east (including NE, E and SE) take up on average

40%, 77%, 39% and 14% of all the trajectory directions during spring, summer, autumn and winter, respectively. Air-mass trajectories originating from the far north bending to the east before their arrival at WLG may catch pollutants if they travel over the large cities. The large occurrence frequency of easterly trajectories in the t=-24h endpoints suggests that anthropogenic influences on WLG should not be neglected. A smaller proportion of the NW trajectories in the t=-24h endpoints was also



detected compared to the t=-168h occurrence frequencies, whereas W trajectories display a larger percentage, indicating that air-masses that originated from the far northwest reach the site by bending to the west of WLG.

It is also worth noting from Figure 2 that the trajectory direction frequencies were far from constant throughout the two decades from 1994 to 2013. There was large interannual variability. Some directions show significant variation trends in their
occurrence frequencies, which will be discussed later on in this section.

The air-mass direction shows that the WLG site is under the influence of different air-masses from different horizontal directions. Apart from that, the WLG site is also under the control of distinct air-masses from different vertical heights throughout the day, with PBL air-masses dominating during the day and FT air-masses during the night, which led to a clear diurnal variation of high nighttime and low daytime ozone concentrations (Ma et al., 2002;Xu et al., 2016). STE events were
also held responsible for the injection of high stratospheric ozone concentrations into the troposphere, leading to elevated surface ozone concentrations (Bonasoni et al., 2000;Ding and Wang, 2006;Stohl et al., 2000;Tang et al., 2011;Lefohn et al., 2012;Jia et al., 2015;Ma et al., 2014;Lee et al., 2007;Liang et al., 2008).

Changes in atmospheric circulations might lead to variations in the occurrence frequencies of air-masses from different directions, which were already discovered in Figure 2. Due to the high dependence of local surface ozone concentrations on
the air-mass origin, a significant change in atmospheric circulations may lead to changing local surface ozone concentrations at WLG. Trajectory directions calculated based on the t=-24h and t=-168h endpoints are used to uncover whether there were secular changes in the occurrence frequency of different directions. Table 1 lists the variation trends (k, slopes of linear regression) of trajectory direction occurrence frequencies in different seasons. The bold numbers in the table are the variation trends that are statistically significant ($\alpha$=0.05). From the t=-168h results it can be noted that, the NW trajectories gained
frequency in the two decades between 1994 and 2013 and the increasing trend is statistically significant in all four seasons. Autumn displays the largest increase followed by spring, respectively showing increasing rates of 1.22 and 1.14 % year$^{-1}$, which would amount to total increases of 24.4% and 22.8% by the end of the two decades. The NW trajectory occurrence frequency increased with a rate of 0.82 % year$^{-1}$ in summer and winter during the two decades, which would amount to an increase of 16.4% in total. As is discussed in the PSCF analysis, the NW trajectories are associated with high ozone
concentrations during all seasons (Figure 1) and thus an increase in its occurrence frequency would therefore lead to an increase in  surface ozone concentration at WLG. The SE trajectories are also often accompanied by high ozone concentrations, representing possible transport of ozone precursors that are of anthropogenic sources from Southeast China and other Southeast Asian countries to WLG. However, the SE trajectories have been significantly decreasing in spring, summer and autumn, with the strongest decrease found in summer (-1.35 % year$^{-1}$), suggesting that changes in air mass origin alone cannot explain the
seasonal ozone trends measured at WLG.

Table 1 shows an increasing trend in the occurrence frequency of E trajectories at t=-24h but not at t=-168h. This indicates that, more trajectories from other directions were turning over to the east of WLG 24h before their arrival at the site. Since the increase in the NW trajectories was only found in the t=-168h trajectories, it is highly possible that the increase of t=-24h E trajectories is associated with the increase in the t=-168h NW trajectories. The SE trajectories at t=-24h, however, show





significant decreasing trends in spring, summer and autumn, with the strongest decrease in summer. This is consistent with the t=-168h results, suggesting that the entire SE air-mass transport pathway decreased in frequency.

Since the NW direction is often linked to high ozone concentrations according to the PSCF and a significant increase in trajectory occurrences from that direction has been detected for t=-168h, a more detailed examination on that part of the
trajectories is highly necessary. After careful examination, it was found that the trajectories starting off in the NW direction mostly bended to the W and E direction or stayed on the NW path. Figure 3 displays the occurrence frequency of trajectories that started off in the NW direction at t=-168h and turned to the E, NW and W direction at t=-24h, with the lines indicating the according decadal linear variation trends. It can be seen that, more and more trajectories bended to the E direction before arriving at WLG, with significant trends in all seasons and a largest increasing slope in summer. The trajectories originating
from the NW and staying on the NW path throughout the transport process take up a relatively smaller proportion compared to the other two pathways and do not show any significant variation trends throughout the two decades except for summer, in which there is a slight decreasing trend (p=0.05). Trajectories turning to the W are most common and they are gaining in frequency in spring, autumn and winter, with autumn showing the largest increasing slope (1% year$^{-1}$). Trajectories staying in the NW and those bending to the W are more likely to keep their original air-mass properties and may therefore show higher
ozone concentrations than the air-masses turning to the E direction. Our previous work (Xu et al., 2016) shows that the largest ozone increase occurs in autumn, followed by spring, and the increasing trend in summer is not significant. This may be partly explained by the fact that the NW trajectory frequency increase is not as large in summer as in spring and autumn and more NW trajectories are turning to the E during summer than in the other seasons.

### 3.2  Impacts of stratospheric exchange versus anthropogenic emission trends

We next examine a suite of GFDL-AM3 simulations designed to isolate the response of ozone to changes in meteorology, stratospheric exchange and anthropogenic emission trends. Figure 4 shows year-to-year variation and long-term trends of observed and modelled ozone at WLG, as well as the modelled stratospheric contribution (O$_3$Strat), for the four seasons over the period 1980-2014. Overall, GFDL-AM3 captures the inter-annual variation of observed surface ozone anomaly, with the correlation coefficient ranging from 0.5 to 0.7 for spring, summer and autumn. The correlations between the observed and
modelled ozone anomaly are significant in all seasons at the 90% confidence level; only the correlation in winter failed the 90% significance test. The modelled ozone trends during 1994 to 2013 are 0.32±0.09 ppbv year$^{-1}$ for spring, 0.25±0.09 ppbv year$^{-1}$ for summer, 0.25±0.10 ppbv year$^{-1}$ for autumn , and 0.13±0.15 ppbv year$^{-1}$ for winter. Compared with the observed ozone trends, the modelled spring, summer and winter trends are slightly overestimated, while the autumn trend is slightly underestimated, but the overall increasing trend is well reproduced by the model. A stratospheric ozone tracer implemented in
GFDL-AM3 (O$_3$Strat; Sect. 2.5) indicates that the stratospheric influence can explain 23% (r=0.48) of the observed ozone interannual variability in spring (Fig.4a) but contributes little to observed variability in other seasons (r<0.1; Fig.4b-d). AM3 O$_3$Strat shows a significant (p<0.05) increasing trend of 0.19±0.16 ppbv year$^{-1}$ over the 1994-2013 period during spring, which





can explain 59% of the simulated and 70% of the observed total surface ozone trend, indicating the importance of STT on raising springtime surface ozone measured at WLG over the past two decades (Fig.4a). The largest stratospheric influences are found in the springs of 1999 and 2012 when $O_3$Strat shows an enhancement coinciding with the observed high-ozone anomaly (Figure 4a). We will further discuss the mechanisms driving these ozone enhancements in Sect. 4. During the other

seasons, in contrast, $O_3$Strat reveals insignificant trends at the 95% confidence level and shows weak correlations with the observed ozone (Fig.4b-d). These results indicate that ozone from STT is one of the major contributors to the modelled increase in springtime ozone at WLG but it cannot explain the observed significant ozone increases in autumn.

To evaluate the effect of pollution transport from Southeast and East Asia, we filter the AM3 BASE simulation with the East

Asian CO tracer (EACOt; see Sect.2.5). Following the approach of Lin et al. (2015b; 2017) for western U.S. sites, we use EACOt to identify days when WLG is strongly influenced by polluted airflow from Southeast Asia (including China) (i.e., EACOt greater than its 67$^{th}$ value during each season). Figure 5 shows the trends of ozone from observations, the BASE simulations and the simulated ozone trends under conditions with strong transport from Southeast Asia ($O_{3,EA}$) for the four seasons. During autumn, the simulated trend of ozone increases to $0.40\pm0.17$ ppbv year$^{-1}$ under the dominant influence from

Southeast Asian air masses, compared to $0.26\pm0.11$ ppbv year$^{-1}$ from the BASE simulation and $0.28\pm0.12$ ppbv year$^{-1}$ from observations (Fig.5c). Similar increases are found for summer when the model is filtered for the East Asian influence (Fig.5b). In contrast, the simulated ozone trend at WLG during spring shows little change from the BASE simulation when filtered for the Southeast Asian influence (Fig.5a), supporting our previous conclusion that the stratospheric influence is an important driver of springtime ozone trends measured at WLG (Fig.4a).

To separate the influences of changes in transport patterns and anthropogenic emission trends, we compare trends of seasonal mean ozone at 700 hPa simulated by GFDL-AM3 with time-varying (BASE) and constant anthropogenic emissions (FIXEMIS) over 1995-2014 (Figure 6). With both emissions and meteorology varying, AM3 BASE simulates increasing free tropospheric ozone trends of as large as 1 ppbv year$^{-1}$ throughout Southeast Asia and Northern Asia for both spring and autumn (Fig.6a and 6c). With emissions held constant in time, AM3 FIXEMIS shows very weak and insignificant ozone trends in Southeast Asia

below 30N latitude (Fig.6b and 6d). During spring, however, FIXEMIS simulates significant ozone increases of 0.2 ppbv yr$^{-1}$ extending from Siberia to Northeastern China and to the subtropical Pacific Ocean (Fig.6b). This finding is consistent with our time-series analysis that increasing ozone in the Northwest flow from STT contributes to raising springtime ozone at WLG (Fig.3a and Fig.4a). During autumn, AM3 shows strong ozone increases across the Asian continent in the BASE simulation but simulates little overall ozone trends (<0.05 ppb yr$^{-1}$ around WLG) in FIXEMIS (Fig.6c versus 6d), indicating that the

observed autumnal ozone increase at WLG reflects the influence from increases in regional anthropogenic precursor emissions in Southeast Asia as opposed to changes in air mass origin.

In summary, the AM3 modelling results clearly show that the spring and autumn increases in WLG surface ozone are governed by different processes. Observed increases in springtime ozone at WLG over the 1994-2013 period are linked to decadal variability in stratospheric ozone input in the northwest airflow, while the autumnal increase of ozone at WLG results





from pollution transport from Southeast Asia, where NOx emissions have increased markedly over the last two decades. Notably, surface ozone increases during autumn in AM3 BASE are most pronounced over the subtropical southeast Asian regions (south of 35ºN; Fig.7a), consistent with the observed surface ozone increase at Hong Kong in South China during autumn (Wang et al., 2009). The model shows somewhat decreasing surface ozone trends in the North China Plain during

autumn (Fig.7a), consistent with observations at Shangdianzi near Beijing and Linan near Shanghai (data not shown), indicating a $NO_x$-saturated ozone production. This north-to-south transition from $NO_x$-saturated to $NO_x$-sensitive $O_3$ production regimes during non-summer seasons has also been observed over the eastern U.S. (Lin et al., 2017). Increasing ozone produced from regional anthropogenic emissions in Southeast Asia during autumn is lofted into the free troposphere via deep convection and mid-latitude storms and is further transported in southern and southwesterly airflow towards WLG (Fig.

7b). This interpretation is consistent with one previous modeling study showing that WLG is more heavily influenced by pollution transported from Southeast Asia in autumn than in spring (Liu et al., 2002).

### 3.3 Impacts of ozone transport versus precursor emissions

Ozone is a secondary air pollutant and has a lifetime of several weeks in the free troposphere, which make it complicated to

explain the origin of the measured ozone concentration. Local photochemical production can enhance $O_3$ at WLG, if higher levels of $O_3$ precursors are transported to the site. However, due to the nonlinear relationship of ozone to $NO_x$ and VOC emissions, $O_3$ can be titrated under high-$NO_x$ conditions, resulting in a decrease in concentration. Estimates of the chemical budget suggest that $O_3$ at WLG was net destroyed under the conditions in July of 1996 (Ma et al., 2002). $O_3$ levels in eastern China are high and have been on the rise during the past two decades (Ding et al., 2008;Wang et al., 2012;Ma et al., 2016;Xu

et al., 2008;Wang et al., 2009b;Wang et al., 2017b), hence there the direct transport of ozone plumes within the troposphere will also have an impact on the ozone trend at WLG. In this section the impacts of direct ozone transport versus that of ozone precursors are discussed.

The impact of direct tropospheric ozone transport on ozone trends at WLG are studied by combining the 3-D TOST data from (Liu et al., 2013) with the back trajectory analysis results (section 2.4). The seasonal average distribution of ozone contribution

to WLG through direct tropospheric transport during 1994-2013 is shown in Fig. 8. It can be noted that the distribution varies with season. Spring shows a major contribution from the western edge of the Tibetan Plateau and a small contribution from Central China to the east of WLG (Fig. 8a). Large contributions from the northwestern to the eastern sector are found in summer, including contributions from Mongolia, Inner Mongolia, Central and Eastern China, where high ozone levels can be observed during summertime. Autumn is strongly influenced by transport from Central and Eastern China and less by the NW

sector, while winter is under the strong influence of transport form the western sector.

The trends of the ozone transport contribution in different seasons were calculated and are depicted in Fig. 8. Spring shows a significantly increasing contribution from the north of WLG and significantly decreasing contribution from the western sector,





where the contribution in spring is largest (see Fig. 8a). Statistically significant increases with small slopes were found in Central Asia and East Europe during summer and winter (Fig.9b, d). Significant increasing trends in ozone contribution with relatively large slopes (>0.5 ppbv year$^{-1}$) can be seen in Central and Eastern China during autumn, while slower increases exist in the western, north western and northern sectors.

5    The total contribution of direct ozone transport to WLG ozone concentration can be calculated for each month and the total and seasonal trend is calculated and listed in Table 2. The overall trend (0.28±0.30 ppbv year$^{-1}$) and that in spring (0.27±0.30 ppbv year$^{-1}$) and summer (0.16±1.11 ppbv year$^{-1}$) agree well with the observed trends at WLG (Xu et al., 2016), however, they are statistically insignificant at (α=0.1). The autumn trend (0.56±0.54 ppbv year$^{-1}$) is much larger than the observed trend (0.28±0.11 ppbv year$^{-1}$) and has passed the 90% significance test, indicating that tropospheric ozone transport has significantly elevated the level of autumn ozone at WLG. No trend was detected in winter, indicating that the ozone trend observed at WLG during winter was not caused by tropospheric ozone transport.

Influences from increasing anthropogenic precursor emissions should mostly come from the east of WLG. In Sect 3.1, an increase in trajectories originating in the NW and turning to the E has been found, while the SE trajectories significantly decreased. Under these circumstances, how does the rapid economic development and increased ozone precursor emissions of the cities to the east of WLG influence its ozone level?

Since CO has a relatively longer lifetime among the primary trace gas pollutants, it can be used to check whether WLG is influenced by the increasing anthropogenic emissions to its east. Figure 10 displays the variation trend of observed seasonal average surface CO at WLG during spring, summer, autumn and winter, respectively. A statistically significant trend could only be found in summer, where CO shows a linear increasing slope of 1.07 ppbv year$^{-1}$. Since summer is the season mostly influenced by easterly air-masses, the rising CO level is most likely the result of growing anthropogenic emissions in the regions to the east of WLG. Using CO as a tracer for anthropogenic pollution, it is clear that WLG is to a certain extent influenced by the growing primary air pollutant emissions to its east.

To investigate the relative importance of the influence from the growing emissions east of WLG, the trajectories and the associated ozone concentrations were grouped into four groups (SW, NW, NE and SE) according to the t=-24h trajectory directions. The trends and the according confidence intervals of the monthly average ozone concentrations associated with different air-mass origins were calculated using the seasonal Mann-Kendall trend analysis and listed in Table 3. It can be noted that, the ozone associated with all trajectory directions showed statistically significant upward trends during 1994 to 2013 at a confidence level of 95%. The eastern sectors (NE and SE) display larger slopes than the western sectors (NW and SW). The largest ozone trend was associated with the SE direction, reaching 0.29 (0.21-0.36), 0.35 (0.24-0.43), 0.29 (0.19-0.39) ppbv year$^{-1}$ respectively for the all-day, daytime and nighttime data subsets. The smallest ozone trend was associated with the NW direction, reaching 0.13(0.07-0.18), 0.14 (0.09-0.19), 0.14 (0.07-0.20) ppbv year$^{-1}$ respectively for the all-day, daytime and nighttime data subsets. This indicates that easterly trajectories, which are more likely to be influenced by anthropogenic emissions of ozone precursors, are associated with larger ozone trend than westerly trajectories, confirming that anthropogenic influence from the east is also leading to the increase of ozone at WLG.





In all, the increase in ozone at WLG is both influenced by direct tropospheric ozone transport and rising precursor emissions in the eastern sector. The increase in direct transport of ozone to WLG only led to a significant rise in autumnal ozone, which supports the conclusions from the modelling study in Sect. 3.2.

## 4  Atmospheric dynamics and ozone variability at WLG

### 4.1  Stratosphere-to-troposphere transport and jet characteristics

The highest ozone concentrations at WLG during spring were observed in 1999 and 2012, coincided with the largest stratospheric influences simulated in the GFDL-AM3 model (Fig.4a). In contrast, the springs of 1998 and 2007 experienced lower observed ozone and simulated stratospheric influence. In this section we investigate the links of these ozone anomalies to changes in the structure of the jet stream. The top panels of Figure 11 show time series of observed daily surface ozone and

modelled $O_3$Strat at WLG from March to May in 1999 and 2012, with the STT ozone transport events marked as the pink shades. During STT events, peaks are found both in observed surface ozone and modelled $O_3$Strat, accompanied mostly by enhancements in PV and ozone in the ERA-interim data, as illustrated for March 30, 2012 (Figure 12). A low pressure system sit over Northeast China on March 30, 2012 (Figure 12a). WLG observatory was located ahead of the cold front. Elevated 250hPa PV is visible (Figure 12b), reaching up to 7 PVU. The midlatitude jet stream (U wind >35 m s$^{-1}$; white dots in Fig.12c)

extended up to 33°N, with a tropopause fold found to its north, directly influencing the STE process at WLG. The cross section of ozone from ECMWF shown in in Figure 12d displays a similar contour shape to the PV cross section, indicating downward intrusions of stratospheric ozone into the troposphere.

Analysis of 200 hPa zonal wind anomalies from the NCEP reanalysis indicates strengthening of the midlatitude jet stream

across the Tibetan Plateau during the springs of 1999 and 2012, with the centre of the jet stream shifted to the north towards WLG compared to the 1994-2013 mean state (Fig.11b). These circulation anomalies facilitate the formation of tropopause folding and transport of stratospheric ozone into the free troposphere above WLG, consistent with frequent STT events as identified by GFDL-AM3 $O_3$Strat (Fig.11a). For comparison, the strength of the jet stream across the Tibetan Plateau was weakened during the springs of 1998 and 2007, leading to weaker stratospheric influence at WLG. In particular, the location

of the subtropical jet was shifted to the south far away from WLG in spring 1998 following a strong El Nino winter. These interpretations are consistent with the findings of Lin et al. (2015a), who showed frequent stratospheric intrusions and high surface ozone events during the springs of 1999 and 2012 when the polar jet stream was unusually contorted over the western United States.





### 4.2 Modes of atmospheric circulation

There are many oscillations within the atmospheric circulation with different periodicities, e.g. QBO with a quasi-2-year periodicity and ENSO with a 2 to 7-year periodicity. Similar periodicities were also found in the surface ozone data at WLG in (Xu et al., 2016). In this section, the potential impacts of different atmospheric circulation oscillations on surface ozone

concentrations at WLG are investigated by comparing climate indices created for the atmospheric circulation oscillations with various IMFs from the EMD analysis in (Xu et al., 2016). Since nighttime ozone concentrations at WLG are more representative of the free tropospheric air condition, IMFs of the nighttime ozone data are applied in the following analysis. Previous studies concluded that tropospheric column ozone over the Tibetan Plateau bears a QBO signal with the same phase as the tropical stratospheric wind QBO, which is caused by the increase and decrease in tropopause height over the Plateau

region, as the tropical stratospheric winds shift from easterly to westerly (Ji et al., 2001). The 3$^{rd}$ IMF of the nighttime surface ozone data reveals a periodicity closest to that of the QBO index. The comparison between the QBO index and the 3$^{rd}$ IMF is displayed in Figure 13. It can be discerned that, the normalized 3$^{rd}$ IMF and the QBO index show a significant positive correlation during nighttime (r=0.21). The peaks and valleys coincide well with each other during 1994-2002 and 2011-2013. During 2003-2010, the QBO index displays 4 peaks, while the 3$^{rd}$ IMF only shows two peaks. This confirms that the surface

ozone is influenced by the tropical stratospheric wind QBO, however, the EMD analysis was not fully able to extract the QBO signal during 2003-2010 probably due to the interference of other signals. During 1994-2013, the QBO influence was able to cause ozone concentration fluctuations in the range of ±1.7 ppbv.

The EASM, ENSO and NAO are other circulation-related factors that might influence surface ozone at WLG through the change of the precipitation or the STE processes. However, these influencing factors are often coupled with each other and a

direct relationship between surface ozone observations and these factors might be hard to determine.

The correlation coefficients between surface ozone concentrations, precipitation and the EASM index (EASMI) for June, July and August is listed in Table 4. Only during July, a significant negative correlation (r=-0.59, significant at a 99% confidence level) could be detected between the ozone concentration and the EASMI, which coincides with the significant correlation between the precipitation rate and the EASMI (r=-0.47, significant at a 95% confidence level). However, no significant

relationship was found between ozone and precipitation, indicating that the precipitation might not be the only or the most dominant process through which the EASM influences ozone.

To investigate the possible impact of the EASM on the STT processes at WLG, the average location of the subtropical jetstream for the top and bottom 15 percentile EASMI cases and the correlation coefficients between the 200hPa zonal wind and the EASMI in June, July and Augustfrom1990 to 2015 were calculated and are displayed in Figure 14a1-c1). A significant positive

correlation band between the zonal wind and the EASMI is found over WLG during June and negative correlation exists to its south and north, indicating that during strong EASM years, the subtropical jetstream shifts to the south towards WLG. In July and August, however, the correlation is negative over WLG and positive to its south and north, indicating that the subtropical jetstream shifts north away from WLG during strong EASM years and shifts to the south towards WLG during weak ones.



The shift in subtropical jetstream location leads to changes in STT, which is confirmed by the simulations results of stratospheric contribution (Table 5). The modelled $O_3$Strat during weak monsoon years are 10%, 19% and 27% higher than those during strong monsoon years in June, July and August, respectively.

The EASM can also change the atmospheric circulation and thus change transport processes over WLG. The average 500 hPa geopotential height distribution is shown in Figure 14a2-c2 and the location where the geopotential height is significantly correlated to the EASMI is marked by + and – signs according to the sign of the correlation coefficients. It can be seen that, WLG is located behind a ridge, which is why WLG is often governed by northwesterly air flows. The center of the western Pacific subtropical high pressure belt shifts northwards during June to August. At the same time a strong low pressure system forms over India and reaches its strongest state in July. The location of the convergence belt between the Indian low and the subtropical high seems to be most in favour of transport from eastern China during July. Strong negative correlation exists between the EASMI and the 500hPa geopotential height to the south of WLG during June to August, while in July the negative correlation exists mostly to the east of the Indian low. During July in weak monsoon years, the subtropical high is enhanced, without weakening the Indian low, which probably is why an increased concentration in ozone associated with high EACOt (6%) is observed during weak monsoon cases in comparison to strong monsoon cases during July. Additionally, increased ozone concentrations are also observed in those associated with high NACOt (4%) during July. June and August both display decreased ozone concentrations associated with the high EACOt and NACOt during weak monsoon years.

In all, weak monsoon years are in favour of the STT ozone transport, especially during July and August, and the circulation pattern during weak monsoon years favours the horizontal transport of ozone to WLG during July, which resulted in the overall effect of strong negative correlation between the EASMI and the ozone concentrations in July. These results are in contradiction with the modelling study by (Yang et al., 2014), which suggested that ozone concentrations were positively correlated to the EASMI.

Past studies have pointed out that the EASM exhibits an enhanced relationship with ENSO(Wang et al., 2009a). The NAO has possible effects on strengthening this EASM-ENSO relationship(Wu et al., 2012) and also displays inter-annual tele-connections with the EASM(Linderholm et al., 2011). Liu and Yin (2001) suggest that in the north-eastern Tibetan Plateau, years with high NAO index values are associated with above-normal summer precipitation, while years with low NAO index values are experiencing below-normal summer precipitation. The correlation between the NAO index and the 3rd+4thIMF of nighttime surface ozone is displayed in Figure 15. A positive correlation (r=0.26, significant at a 99% confidence level) is detected and the combination of the 3rd and 4thIMF captures well the fluctuations in NAO index. A lead-lag correlation analysis between the NAO index and the 3rd+4thIMF displays a maximum correlation (r=0.28) at a lead of the NAO index by three months. This suggests that high NAO indices are causing positive ozone anomalies, delayed by threemonths, while low NAO indices lead to negative ozone anomalies. The mechanism behind such a teleconnection is so far not clear and needs further exploration. The fluctuations in NAO index can lead to ozone anomalies within the range of ±3.0 ppbv.



### 4.3 The impact of solar activities

A solar cycle signal was found in the tropospheric ozone column data over the Tibetan Plateau (Huang et al., 2009). Here, we investigate the impact of solar activities on surface ozone trends at WLG by comparing the normalized 1-year running average SSN with the normalized daytime and nighttime 5$^{th}$ intrinsic mode function (IMF) of monthly average ozone that were obtained
in our previous study (Xu et al., 2016).

Results are displayed in Figure 16, which reveals that both the daytime and nighttime 5$^{th}$ IMF revealed significant positive correlation to the SSN, with daytime (r=0.70) showing a better correlation than nighttime (r=0.43). During the 1994-2013 period, there were two valleys of the SSN respectively in 1996 and 2008 and two broad peaks respectively during 2000-2002 and 2012-2014. The occurrence time of the two valleys in the daytime 5$^{th}$ IMF agrees well with that of the SSN, while the first
peak shows a delay of 1-2 years. The nighttime 5$^{th}$ IMF displays both delayed valley in 1997 and peak in 2004.

The positive correlation between the 5$^{th}$ IMF and the SSN explains the 11-year periodicity found in the ozone data. Solar activity led to surface ozone variations within the range of ±0.5 ppbv over the period of 1994 to 2013 (see Fig. 6 in Xu et al., 2016).

### 5   Multivariate regression of surface ozone at Waliguan

The above analysis suggests that surface ozone at WLG can be influenced by various factors. Some of these factors mainly disturbed the seasonal variation of ozone and contributed to the inter-annual differences, others contributed also to the observed long-term trends. To quantify the contributions of different factors to surface ozone at WLG, a multivariate regression was performed, with normalized monthly ozone concentration being dependent and time and the potential influencing factors being independent variables. All candidate independent variables, e.g., the QBO index, the NAO index, the SSN, the modelled
O$_3$Strat, the NW trajectory frequency (f(NW)), the SE trajectory frequency (f(SE)), and the calculated direct transport contribution of tropospheric ozone (O$_{3,trop}$), were converted to normalized monthly values. The regression equation takes the form described in Sect. 2.6.

The regression was conducted stepwise to avoid overfitting. The coefficient of determination (R$^2$) and residual sum of squares (RSS) were calculated after each step of the regression. Correlation coefficients were calculated between the residual and all
remaining variables. The variable that correlates best with the residual was chosen as the next independent variable to be included in the model. The regression stopped when the changes in R$^2$ and RSS were less than 1%. The first step of the regression was to fit the third order harmonic function (6) to the normalized ozone data. Five factors (i.e., O$_3$Strat, O$_{3,trop}$, SSN, f(NW), and QBO index) were successively included in the regression and became independent variables. Changes of R$^2$ and RSS after each step are shown in the supplement (Fig. S1). The regression coefficients are listed in Table 6. An empirical
model for normalized monthly ozone at WLG is obtained by integrating the regression coefficients in Table 6 into equations (5)-(7). This empirical model is used for the calculation of normalized monthly ozone at the site.





Figure 17 shows a comparison between the calculated and observed ozone, together with the calculated contributions of the influencing factors to the normalized monthly ozone at the site. It can be seen that the calculated normalized ozone reproduces well the observed one ($R^2$=0.92). The differences (residual) between the observed and calculated normalized ozone are within ±0.25 and mostly within ±0.10. An ozone trend of 0.25 ppbv year$^{-1}$ is obtained from the observational data, while the calculated

ozone data gives a trend of only 0.08 ppbv year$^{-1}$. The discrepancy can partly be explained by the trend in the residual (0.11 ppbv year$^{-1}$). The rest should be due to the uncertainties associated with the empirical model as well as the independents. The regression produced a background signal in the normalized ozone, with amplitude of about 0.67 and no trend. The modelled $O_3$Strat, $O_{3,trop}$, SSN, f(NW) and QBO contribute up to 0.32, 0.17, 0.11, 0.12 and 0.04, respectively, to the calculated normalized ozone. These results indicate that the level of surface ozone at WLG has a basic component (the background signal),

which makes the major contribution to the seasonal variation of ozone but has no long-term trend. The background signal is enhanced by varying contributions from STT, tropospheric ozone transport and sunspot number, influenced by changes in the NW trajectory frequencies, and very little by QBO.

## 6    Conclusions

Through an observational and modelling analysis, we have discussed the key drivers of various periodicities and long-term

trends of ozone measured at WLG for the four seasons over the past two decades, previously reported in the companion paper (Xu et al., 2016). The impact of air mass origin is investigated using backward trajectory analysis combined with PSCF analysis, the influence of STE and increasing anthropogenic emissions in Asia is evaluated using chemistry-climate model hindcasts driven by reanalysis winds (GFDL-AM3; Lin et al., 2017). The impact of direct tropospheric ozone transport on ozone at WLG is examined using 3D tropospheric ozone climatology data (a subset of TOST) combined with the trajectory analysis results.

Our result show that different processes have contributed to the observed increasing ozone trends at WLG during spring versus autumn. Analysis of a stratospheric ozone tracer in GFDL-AM3 indicates that STT can explain ~60% of the simulated and ~70% of the total observed springtime ozone increase over 1994-2013 at WLG (Fig.4a). This interpretation is consistent with an increase in the NW air mass frequency over this period inferred from the trajectory analysis (Fig.3). STT contributes to the

observed high-ozone anomalies at WLG during the springs of 1999 and 2012 (Figs 10 and 11), linked to the unusual structure of the jet stream as occurs over the western United States during the same years (Lin et al., 2015a). During autumn, observations at WLG are more heavily influenced by polluted air masses originated from Southeast Asia than in the other seasons (Fig.1 and Fig.7). The GFDL-AM3 model captures the observed ozone increase at WLG during autumn (0.26±0.11 ppbv year$^{-1}$) and simulates a greater ozone increase of 0.38±0.11 ppbv year$^{-1}$ under conditions with strong transport from

Southeast Asia (Fig.5c), indicating that rising anthropogenic emissions of ozone precursors in Southeast Asia play a key role in raising ozone observed at WLG during autumn (Fig.6). During summer, WLG is mostly influenced by easterly air masses from the cities to the east of WLG but these trajectories these trajectories do not extend to the polluted regions of eastern China





and have decreased significantly over the last two decades (Fig.2), which likely explains why summertime ozone measured at WLG shows no significant trend despite ozone increases in Eastern China. The direct transport of tropospheric ozone to WLG calculated from the TOST data and trajectory residence time reveals significant increases during autumn mostly coming from the eastern sector (Tab. 2 and Figs. 8 and 9).

The periodicities detected in the HHT analysis of ozone data previously reported by Xu et al. (2016) is linked to various climate indices including EASMI, NAO and sunspot cycle. The 2-3 year periodicity is linked to the QBO and the 3-7 year periodicity could be partly explained by the EASMI and NAO, while the 11 year periodicity is well connected to the sunspot cycle. An empirical model is obtained for normalized monthly level of surface ozone at WLG using the multivariate regression technique

and used to explain the observed ozone trends. Based on these relationships, an empirical model has been established for normalized monthly ozone through multivariate regression. The regression model reproduces well the observation and can capture about one third of the observed ozone trend.

The results obtained in this work clearly show the complexity of surface ozone in terms of influencing factors. Comprehensive investigations are recommended to understand variations of surface ozone at any sites, in particular the long-term trends. Our

work in this paper and the companion paper shows an example of de-convoluting the ozone variations and interpreting those using related dynamical and chemical factors of different scales, which hopefully can inspire similar studies.

**Data availability**

The ozone data analysed in this work are partly available at the World Data Center for Greenhouse Gases(WDCGG) (http://ds.data.jma.go.jp/gmd/wdcgg/cgi-bin/wdcgg/download.cgi?index=WLG236N00-

CMA¶m=201405120001&select=inventory). The entire data set can be made available for scientific purposes upon request to the corresponding author (xuxb@camscma.cn). The AM3 global model simulations are archived at GFDL and are available to the public upon request to Meiyun Lin (Meiyun.Lin@noaa.gov).

**Acknowledgements**

This work is supported by the Natural Science Foundation of China (No.41330422 and 41505107), China Special Fund for Meteorological Research in the Public Interest (No. GYHY201106023), Environmental Protection Public Welfare Scientific Research Project, Ministry of Environmental Protection of the People's Republic of China (No. 201509002), the Key research and development program of the Ministry of science and technology (No. 2016YFC0202300) and the Basic Research Fund of CAMS (No. 2016Y006).





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



Table 1 The linear variation slope (k)of trajectory direction occurrence frequency for t=-24h and -168h (numbers in bold are significant under a confidence level of 95%).

| k (% a⁻¹) | Season | Trajectory direction | | | | | | | |
|---|---|---|---|---|---|---|---|---|---|
| | | S | SW | W | NW | N | NE | E | SE |
| t=-24h | MAM | **-0.17** | -0.06 | 0.32 | -0.33 | -0.01 | -0.15 | **1.11** | **-0.70** |
| | JJA | **-0.31** | -0.13 | -0.04 | -0.20 | -0.04 | -0.07 | **2.87** | **-2.08** |
| | SON | -0.11 | -0.18 | 0.57 | -0.41 | **-0.12** | **-0.32** | **1.43** | **-0.87** |
| | DJF | -0.02 | -0.14 | -0.34 | -0.14 | -0.04 | **-0.17** | **0.85** | -0.01 |
| t=-168h | MAM | 0.01 | 0.00 | 0.11 | **1.14** | -0.26 | 0.03 | **-0.52** | **-0.51** |
| | JJA | -0.09 | **0.06** | **0.64** | **0.82** | 0.19 | 0.26 | -0.53 | **-1.35** |
| | SON | **-0.13** | **-0.44** | -0.05 | **1.22** | 0.02 | 0.04 | -0.21 | **-0.44** |
| | DJF | -0.01 | -0.35 | -0.54 | **0.82** | 0.11 | 0.03 | -0.03 | -0.03 |

Table 2 Trend of total contribution of direct ozone transport on WLG ozone concentration (ppbv year⁻¹), confidence intervals are given for α=0.1

| Season | all-year | Spring | Summer | Autumn | Winter |
|---|---|---|---|---|---|
| Slope | 0.28±0.30 | 0.27±0.30 | 0.16±1.11 | **0.56±0.54** | 0.01±0.32 |
| P-value | 0.12 | 0.13 | 0.80 | **0.09** | 0.98 |



Table 3  The Kendall's variation slope (k, ppbv year$^{-1}$) of ozone concentrations associated with different trajectory directions, the according confidence interval and p-values.

| Variable | Time of day | Trajectory direction | | | |
|---|---|---|---|---|---|
| | | SW | NW | NE | SE |
| k (ppbv year$^{-1}$) | all day | 0.17 | 0.13 | 0.23 | 0.29 |
| | day | 0.17 | 0.14 | 0.29 | 0.35 |
| | night | 0.15 | 0.14 | 0.30 | 0.29 |
| 95% Confidence Interval (ppbv year$^{-1}$) | all day | 0.11-0.24 | 0.07-0.18 | 0.15-0.32 | 0.21-0.36 |
| | day | 0.12-0.25 | 0.09-0.19 | 0.18-0.39 | 0.24-0.43 |
| | night | 0.08-0.23 | 0.07-0.20 | 0.21-0.39 | 0.19-0.39 |
| p | all day | <0.01 | 0.01 | <0.01 | <0.01 |
| | day | 0.01 | <0.01 | <0.01 | <0.01 |
| | night | <0.01 | 0.01 | <0.01 | <0.01 |

Table 4 Correlation between the surface ozone (1994-2013), the NCEP Reanalysis Precipitation (1990-2015) and the EASMI (1990-2015)

| r (p-value) | June | July | August |
|---|---|---|---|
| Precipitation & EASMI | -0.12 (0.15) | **-0.47 (0.02)** | 0.21 (0.12) |
| $O_3$ & EASMI | -0.08 (0.76) | **-0.59 (0.01)** | -0.32 (0.20) |
| $O_3$ & Precipitation | **0.30 (0.10)** | -0.01 (0.20) | **-0.51(0.03)** |



Table 5 The changes in stratospheric ozone input and in ozone concentration associated with East Asian, European and North American transport (%) introduced by the East Asian Monsoon.

| Month | $(O_{3,\,EASMI<=15th}-O_{3,EASMI>=85th})/\bar{O}_3$ (%) | | | | | |
| | $O_3Strat$ | $O_{3,ea}$ | $O_{3,eu}$ | $O_{3,na}$ | Mean | $O_{3,WLG}$ |
|---|---|---|---|---|---|---|
| June | 10.4 | -6.2 | 1.3 | -3.8 | 0.4 | -0.5 |
| July | 18.9 | 6.2 | 0.1 | 4.3 | 7.4 | 8.7 |
| August | 26.6 | -0.4 | -3.2 | -3.2 | 4.9 | 4.5 |

Table 6 Multivariate regression coefficients (Eq. 6-7) of the surface ozone at WLG

| factor | Regression Coefficients | | | | | | |
|---|---|---|---|---|---|---|---|
| t | $t_0$ | | | | | | |
| | 1.003 | | | | | | |
| | $a_0$ | $a_{1,1}$ | $a_{2,1}$ | $a_{1,2}$ | $a_{2,2}$ | $a_{1,3}$ | $a_{2,3}$ |
| BKG | 0.190 | -0.250 | 0.229 | 0.028 | -0.007 | -0.005 | -0.012 |
| | $b_{i,0}$ | $b_{i,1}$ | $b_{i,2}$ | | | | |
| $O_3Strat$ | 0.336 | -0.135 | -0.109 | | | | |
| $O_{3,trop}$ | 0.100 | 0.042 | -0.112 | | | | |
| SSN | 0.057 | -0.031 | -0.055 | | | | |
| $NW_{freq}$ | 0.111 | -0.048 | -0.100 | | | | |
| QBO | 0.021 | -0.006 | -0.018 | | | | |





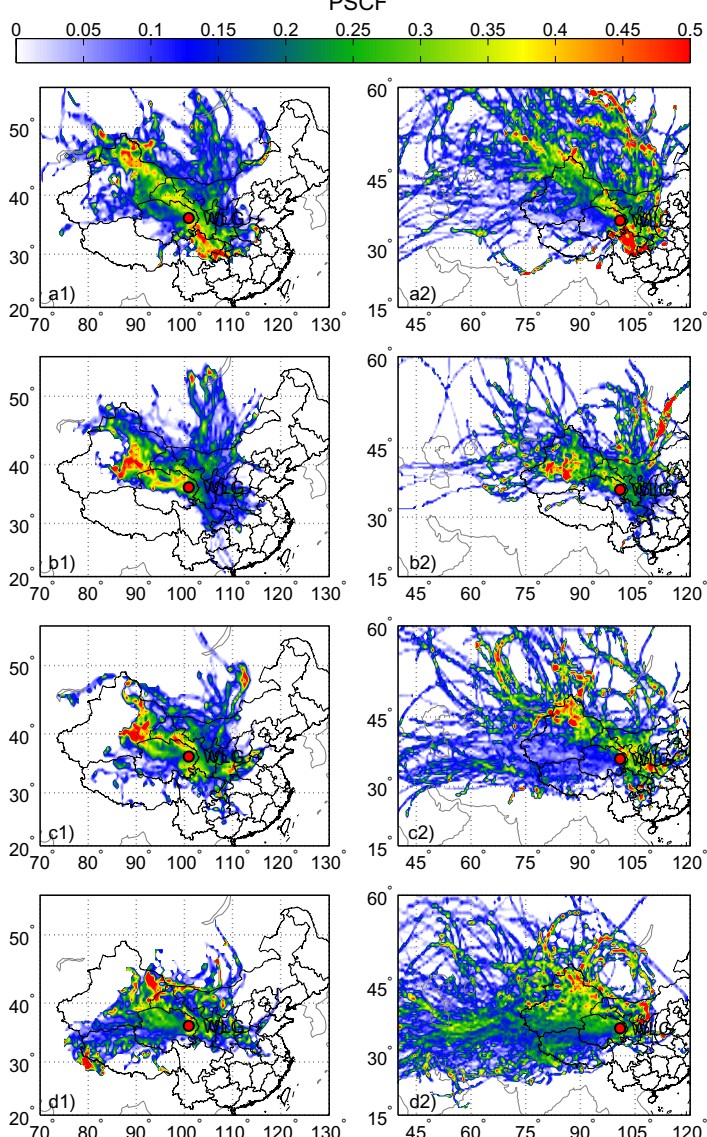

Figure 1 The 1994-2013 climatology of air mass origins at WLG in the PBL (left) and FT (right) for spring (a), summer (b), autumn (c) and winter (d), based on the PSCF analysis (Sect. 2.3).




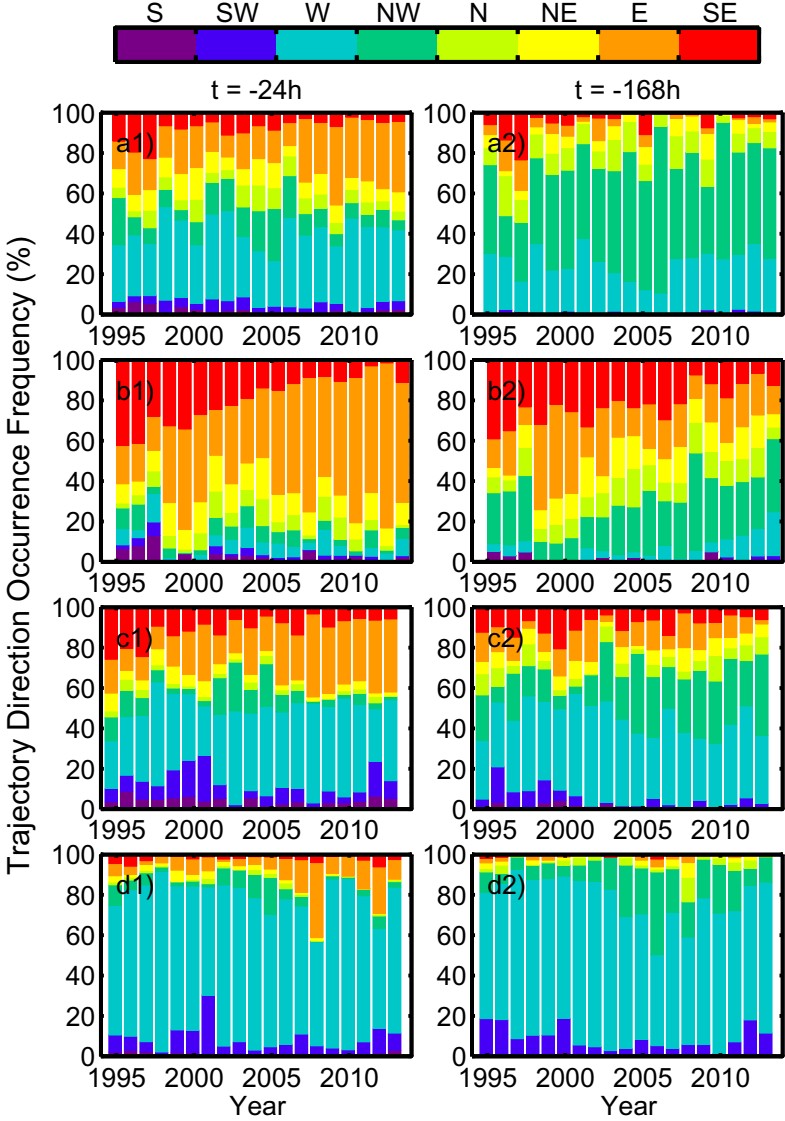

Figure 2 The trajectory direction occurrence frequencies in a) spring, b) summer, c) autumn and d) winter at 1) t=-24h and 2) t=-168h.





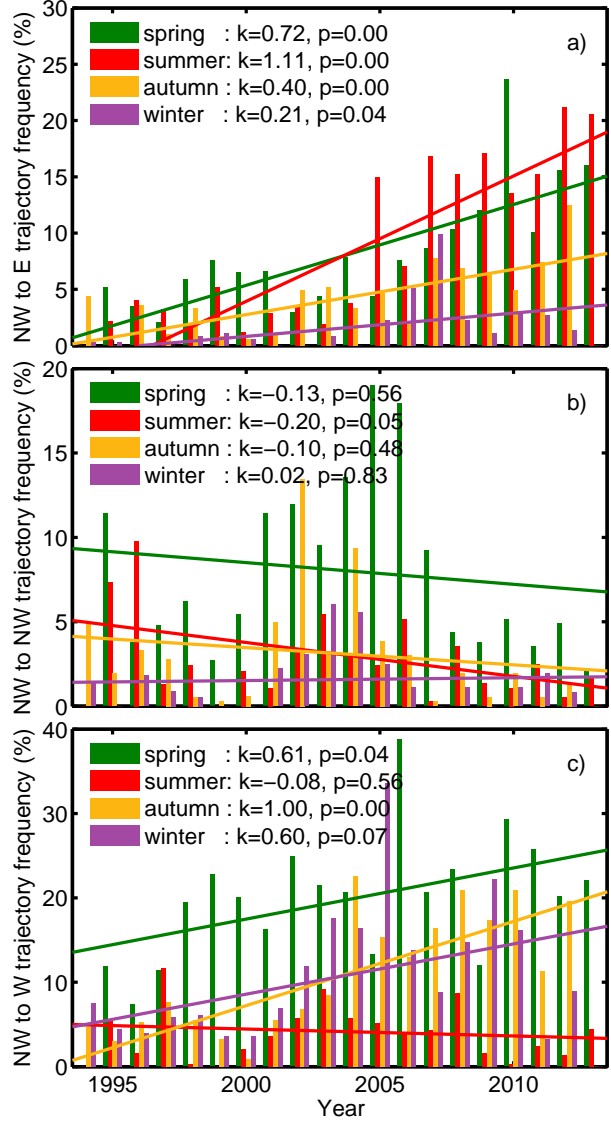

Figure 3 The occurrence frequency of trajectories that originate from the NW at t=-168h and turn to the E (a), NW (b) and W(c) at t=-24h in spring (green), summer (red), autumn (orange) and winter (purple). Bars stand for the occurrence frequencies, while lines are their corresponding linear trends.




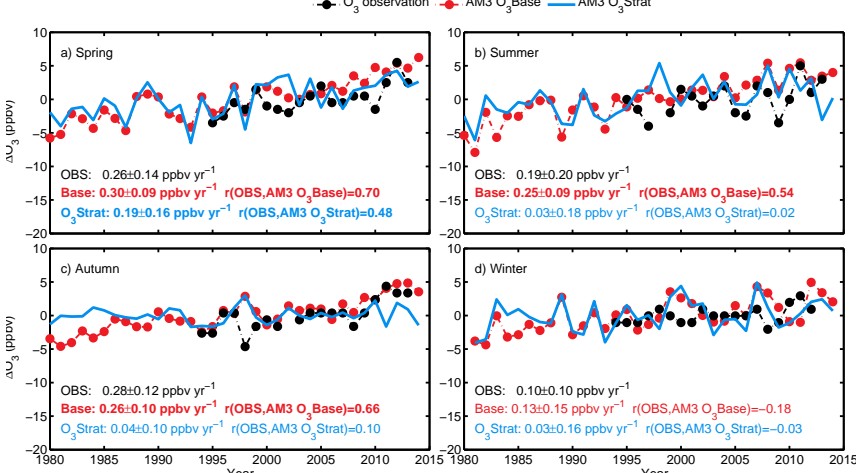

Figure 4 Comparison of seasonal median ozone anomalies at Mt. Waliguan over the period 1980-2014 from available observations (black), GFDL-AM3 BASE simulations (red) and AM3 stratospheric ozone tracer ($O_3$Strat, blue) for a) spring, b) summer, c) autumn and d) winter. The linear trends (with the 95% confidence intervals) over the period 1994-2013 and correlations between observations and models are shown.

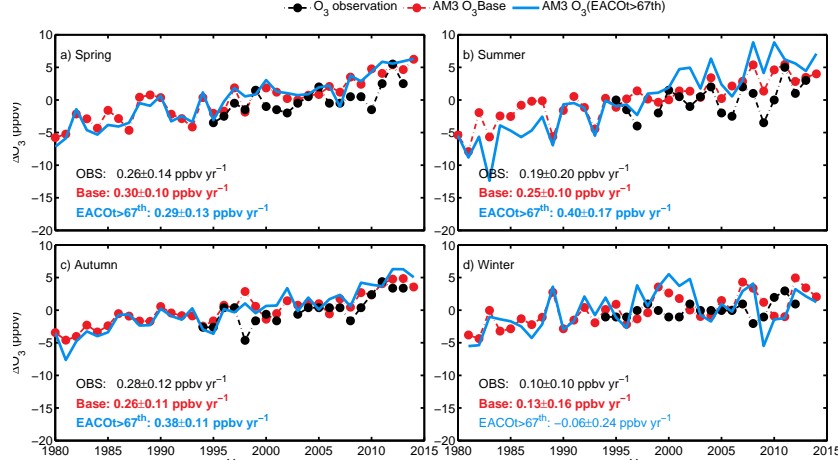

Figure 5 Comparison of seasonal median ozone anomalies at Mt. Waliguan over the period 1980-2014 from available observations (black), GFDL-AM3 BASE simulations (red) and under conditions with strong transport from East Asia (EACOt > 67th, blue) for a) spring, b) summer, c) autumn and d) winter. The linear trends (with the 95% confidence intervals) over the period 1994-2013 and correlations between observations and models are shown.





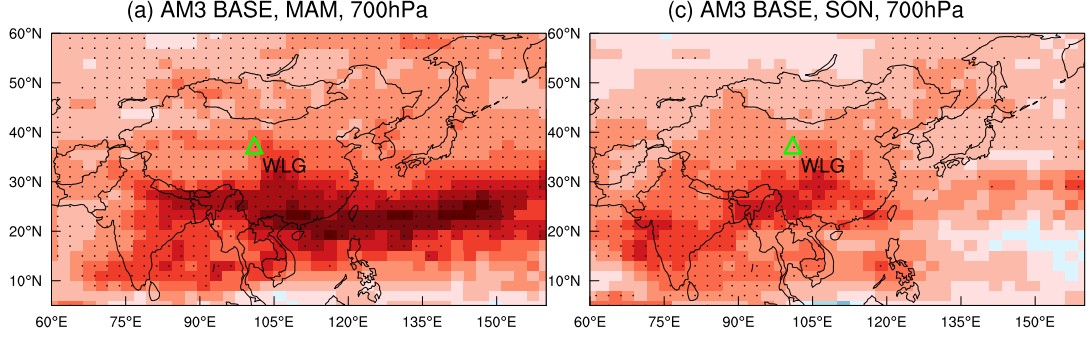

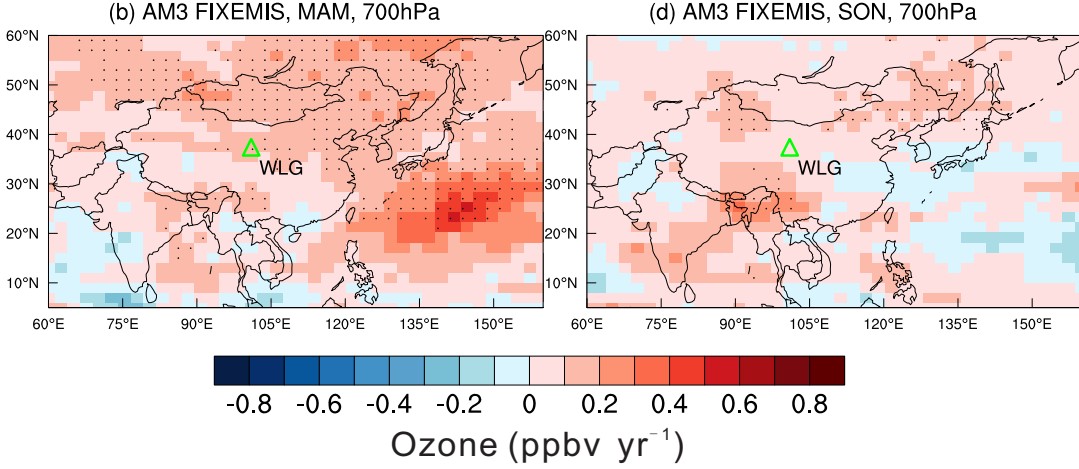

Figure 6 (a-b) The 1995-2014 trends of springtime average ozone sampled at 700 hPa as simulated by the GFDL-AM3 model with time-varying (BASE) and constant anthropogenic emissions (FIXEMIS). (c-d) Same as (a-b) but for autumn. Triangle denote the location of WLG. Stippling indicates areas where the trend is statistically significant at the 95% confidence level.



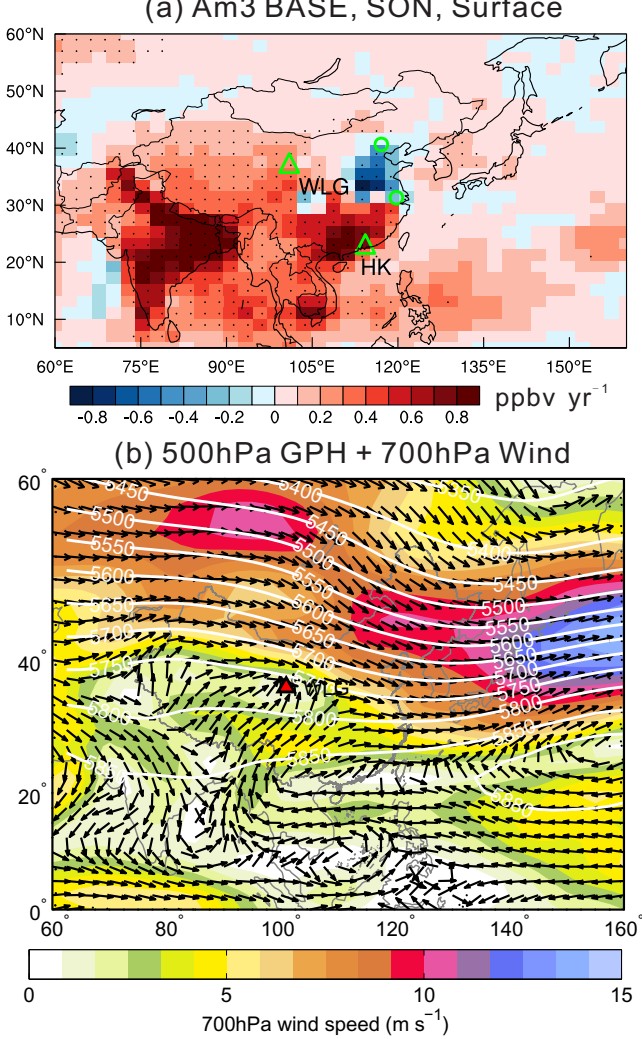

Figure 7 (a) The 1995-2014 trends of autumn average daily maximum 8-hour average ozone sampled in the surface level from the GFDL-AM3 model with time-varying anthropogenic emissions (BASE). Stippling indicates areas where the trend is statistically significant at the 95% confidence level. Green symbols denote the locations of WLG, Hong Kong, Shangdianzi and LinAn. (b) Mean 700hPa wind speed (color shading), direction (black arrows) and 500hPa geopotential height (contours) in autumn averaged over the 1994-2013 period.



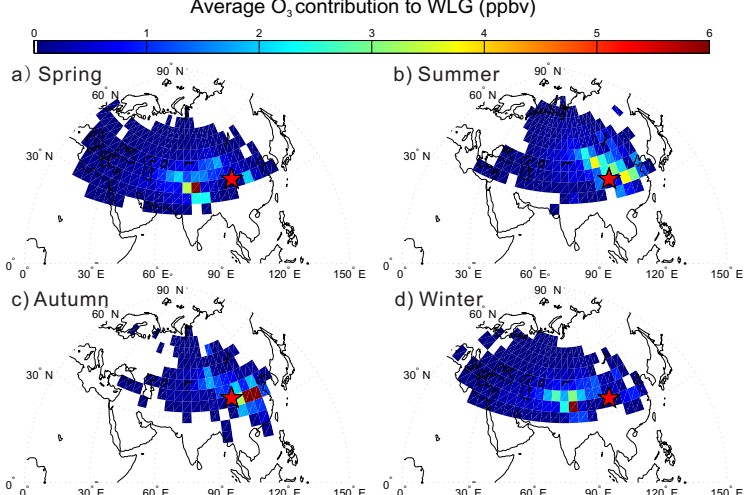

Figure 8 Average seasonal distributions of ozone contribution to WLG through direct transport of ozone during 1994-2013

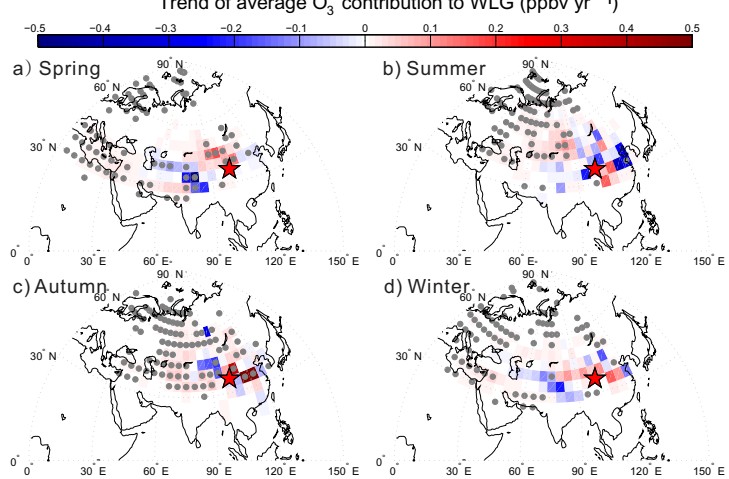

5  Figure 9 Average seasonal distributions of the trend of ozone contribution to WLG through direct transport of ozone during 1994-2013, grey dots stand for the grids passing the 95% confidence test.



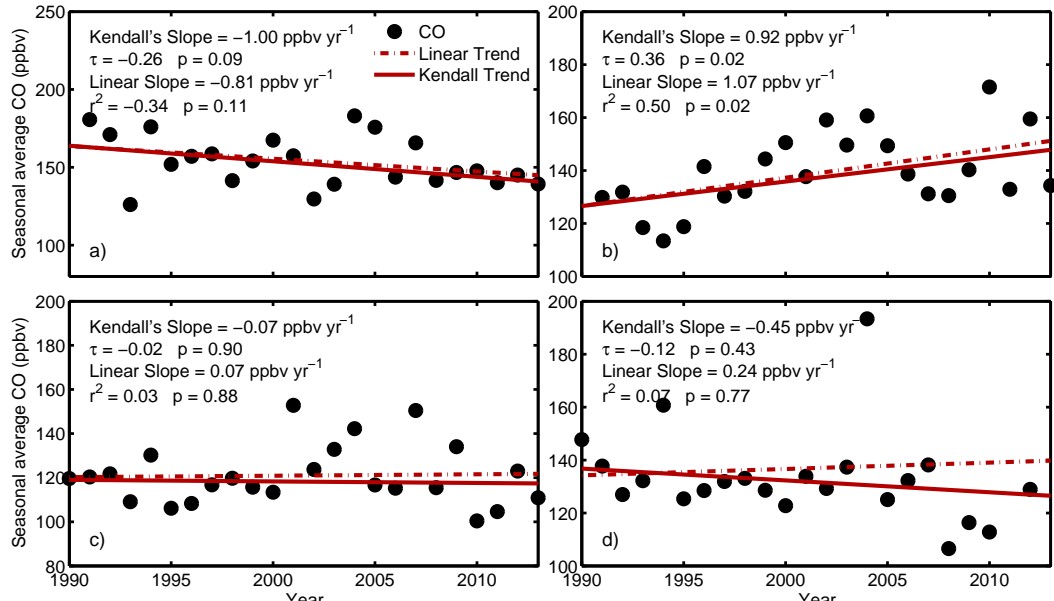

Figure 10 Mann-Kendall and linear trends of seasonal average CO observed during a) spring, b) summer, d) autumn and d) winter from 1990-2013 at WLG





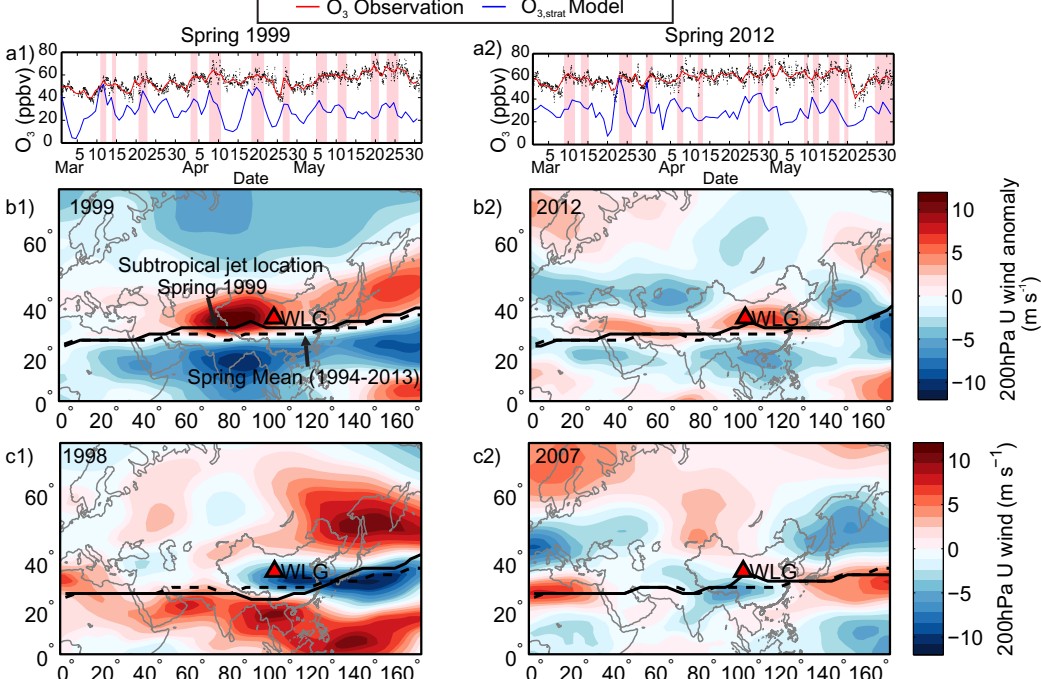

Figure 11 Temporal variations of a) hourly (black dots) and daily (red line) mean surface ozone observations and modelled O₃Strat (blue line) from March to May in 1999 and 2012; b) 200 hPa zonal wind anomaly in the springs of 1999 and 2012 with the strong stratospheric influence; c) Same as (a) but for the springs of 1998 and 2007 with weak stratospheric influence.


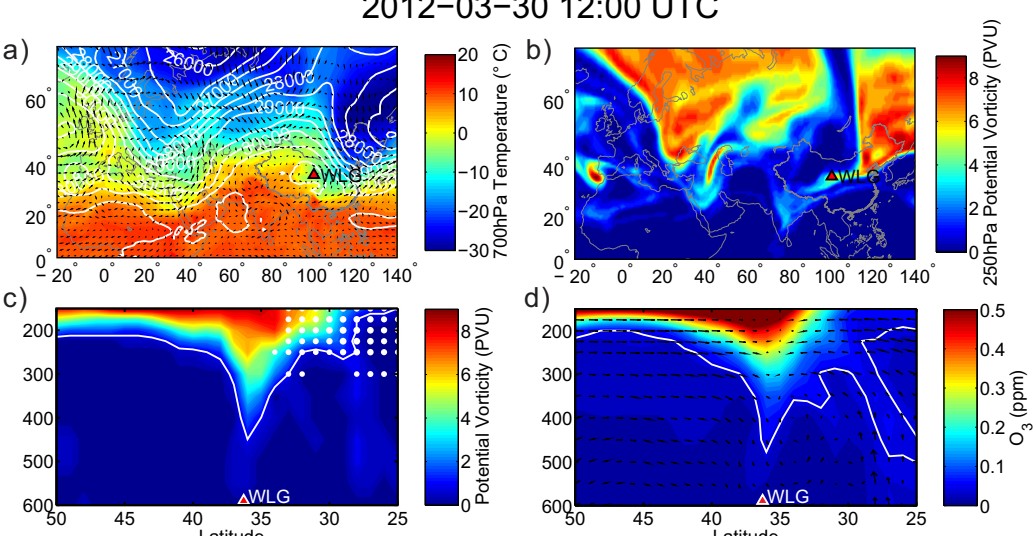

Figure 12a) Map of 700hPa temperature (shading), geopotential height (white contours) and wind field (black arrows); b) Map of 250hPa potential vorticity; c) the cross-section of potential vorticity along the 101.0E longitude line. The white line denotes the 1 PVU isoline and the white dots indicate the location of the subtropical jet stream (U wind>35m s⁻¹); d) The cross-section of ozone mixing ratios along the 101.0E longitude line from the ECMWF reanalysis during an STT transport event on 30 Mar 2012. The while line denotes the 50-ppbv ozone contour.





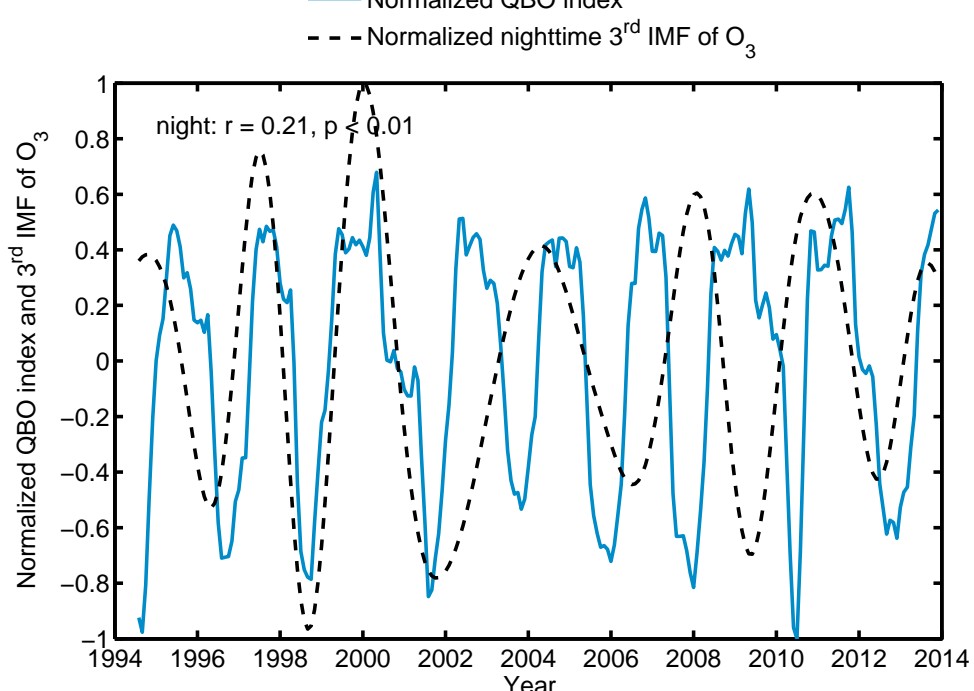

Figure 13 Comparison between the normalized nighttime (dashed black line) 3rd IMF and the normalized QBO index (solid blue line) during 1994-2013.





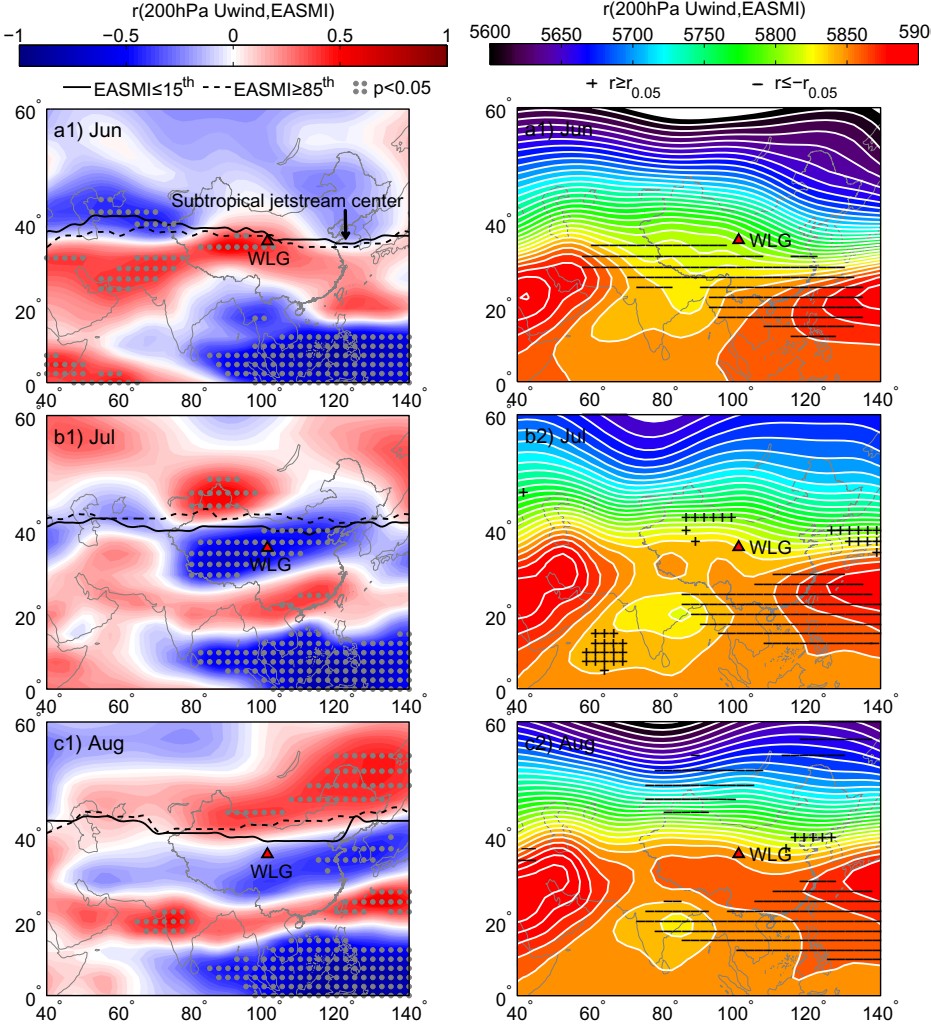

Figure 14 1) The correlation coefficient between the 200hPa zonal wind and the EASMI and the average location of the subtropical jetstream center for the EASMI≤15[th] and EASMI≥85[th] cases and 2) the average 500hPa geopotential height and the location of significant positive(+)/negative(-) correlation (α=0.05) between 500hPa geopotential height and the EASMI in a) Jun, b) Jul and c) Aug during 1990 to 2015.



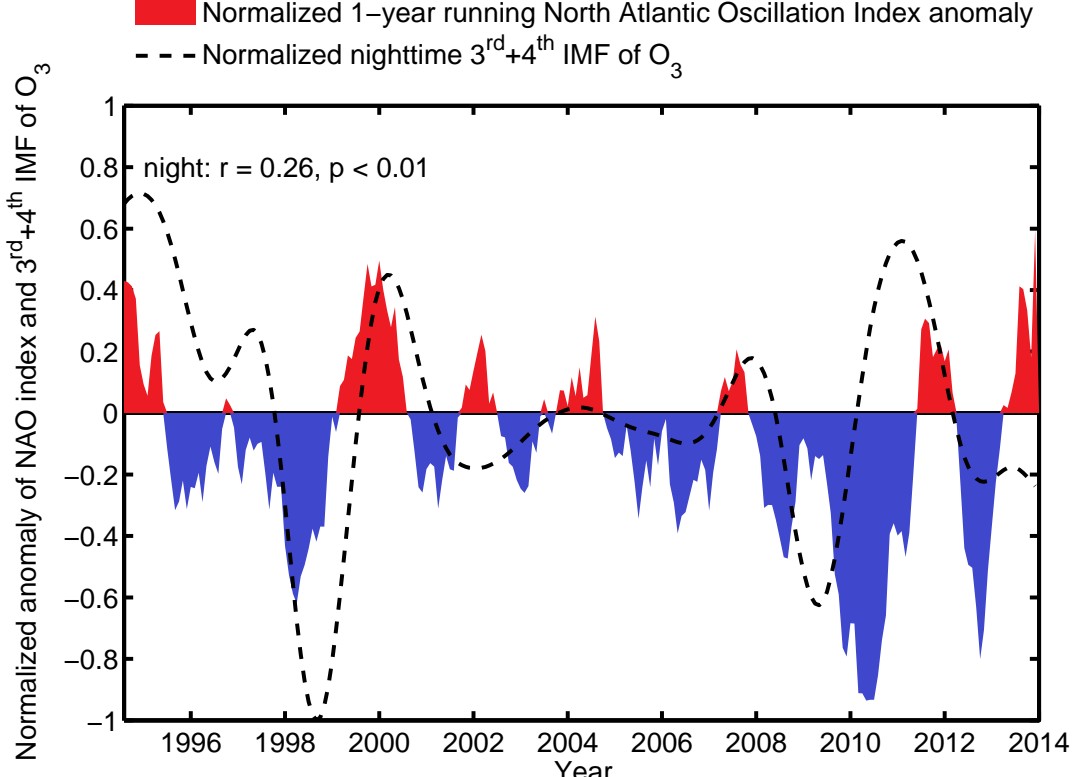

Figure 15 Comparison between the normalized nighttime $3^{rd}$ + $4^{th}$ IMF and the normalized 1-year running average NAO index during 1994-2013.




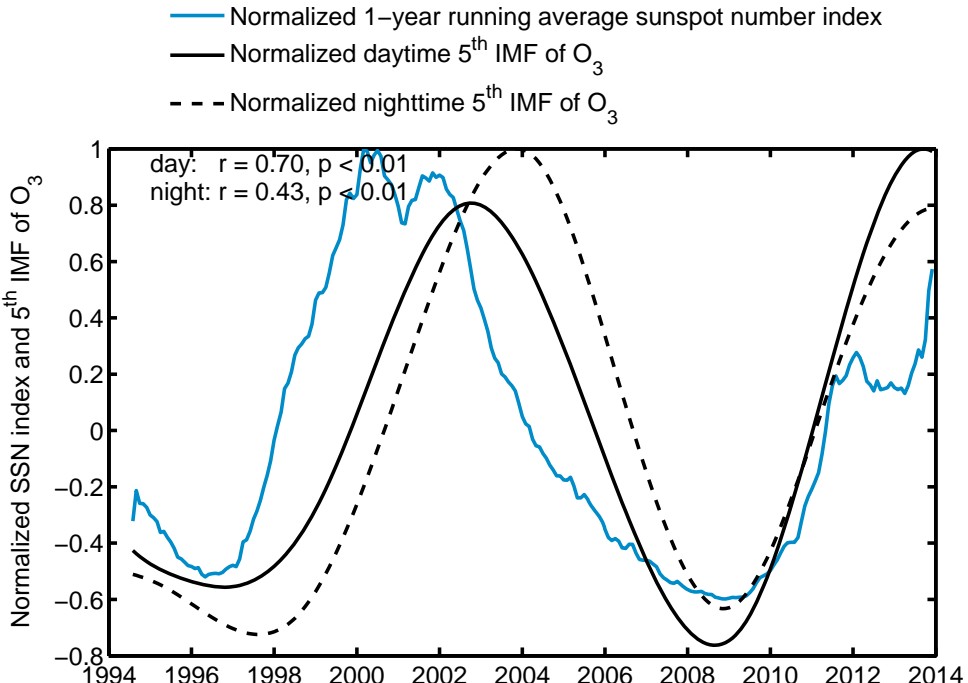

Figure 16 Comparison between the 5th IMF and the 1-year running average SSN during 1994-2013.





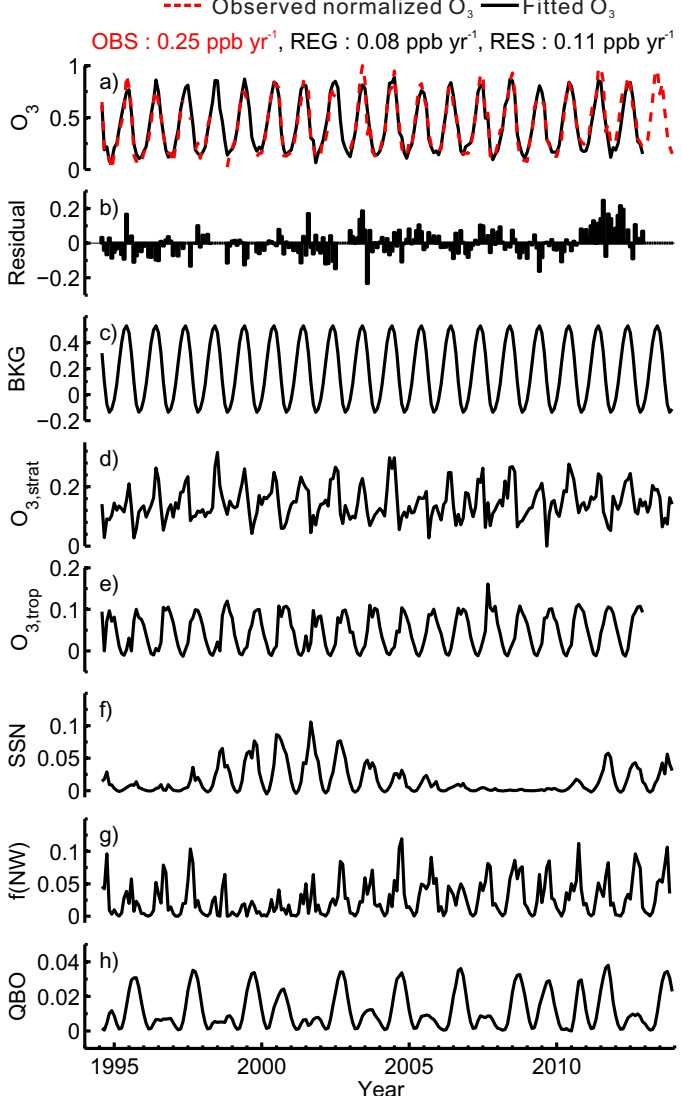

Figure 17 Temporal variation of the a) normalized observed and fitted ozone concentration, b) the residual of the regression, the contribution of c) a background factor and 5 influencing factors to the ozone regression: d) the modelled $O_3Strat$, e) the contribution of tropospheric ozone transport $O_{3,trop}$, f) the sunspot number, g) the frequency of northwesterly trajectories and 5 h) the QBO index