# Peer review of "Figure S1 Changes of the coefficient of determination ( $R^2$ ) and residual sum of squares (RSS) after each step of the multivariate regression."

_Atmospheric Chemistry and Physics, 2017_

## Referee Comment (RC1) · Anonymous Referee #1 · 4 Aug 2017

Long-term trends of surface ozone and its influencing factors at the Mt. Waliguan GAW station, China, Part 2: Variation mechanism and links to some climate indices. Author(s): Wanyun Xu et al. MS No.: acp-2017-483 MS Type: Research article Special Issue: Study of ozone, aerosols and radiation over the Tibetan Plateau (SOAR-TP) (ACP/AMT inter-journal SI)

General comments This paper explores the factors driving the observed ozone changes at Mt. Waliguan Observatory (WLG) using basically backward trajectory analysis and chemistry-climate model hindcast simulations (GFDL-AM3). The paper also

deals links of ozone variability at WLG with the QBO, NAO, the East Asian summer monsoon (EASM), and the sunspot cycle. Although the paper addresses very interesting topics (probably too many issues in a single paper), complementary to that addressed in the companion paper (Xu et al., 2016), in a region of enormous interest such as the Tibetan Plateau, and using valuable data from a global GAW station such as WLG, the paper suffers from significant weaknesses that must be addressed with more credible and robust approaches.

The most important drawbacks are of methodological nature, and are briefly discussed below. Specific comments

1) The approach used for the backtrajectories dataset and climatology does not seem the most advisable to distinguish between ozone long-range transport from ozone produced by regional precursors. The use of the directions of only start-points (origin) of the trajectories into bins of 45° is a very weak approximation. Air masses normally move among sectors along their entire trajectory (especially those of 7 days duration). So, it seems more reasonable to use some index accounting for the time of residence of the trajectory in each geographical sector. Bins of 45° seem to be too narrow for 7-day backtrajectories for which a great error / uncertainty in the geographical determination is associated.

2) The use of 1-day trajectories to estimate the impact of regional ozone sources and those of 7-day path (very long) as representative of ozone long-range transport are not well understood and not sufficiently justified. In fact, when the 1994-2013 climatology of air mass origins at WLG in the PBL and FT are depicted (Figure 1), the main patterns in the distribution of the air massesfrequency is quite similar for both regions (PBL and FT). Indeed, that means that the discrimination between PBL and FT air masses has not been satisfactorily achieved.

3) Nothing is said about the methodology used to determine the critical height of the back-trajectory in relation to the PBL height for each point of the airmass trajectory.

[Figure]

4) In page 8 Lines 5-13; The results are inconsistent and, in some cases, contradictory. Section 3.1 is plenty of inconsistencies such as the following in page 8 lines 16-18:"The t=-168h trajectory direction provides us information on the overall origin of the air-mass, while the trajectory direction calculated for t=-24h should be able to reveal if the air-mass passed over nearby polluted regions before arriving at the station", while in lines 26-27, is said: "From the t=-168h trajectory direction frequencies, it can be seen that the anthropogenic influence is negligible in all seasons"

5) Analyzing 24h and 7-day trajectories, how it is possible to say that "....with PBL air-masses dominating during the day and FT air-masses during the night, which led to a clear diurnal variation of high nighttime and low daytime ozone concentrations". This situation, which is very realistic, probably overturns all the assumptions made for the establishment of the methodology of FT and PBL backtrajectories.

6) All of Section 3.1 should be reviewed using a consistent methodology.

7) In Section 3.2 is difficult to support a joint analysis of point observations in WLG with simulations of GFDL-AM3 with a resolution of 200X200 km2.

8) I do not see GFDL-AM3 captures the inter-annual variation of observed surface ozone anomaly, with the correlation coefficient ranging from 0.5 to 0.7 for spring, summer and autumn, as it is said.

9) The sentence "A stratospheric ozone tracer implemented in 30 GFDL-AM3 (O3Strat; Sect. 2.5) indicates that the stratospheric influence can explain 23% (r=0.48) of the observed ozone interannual variability in spring (Fig.4a) but contributes little to observed variability in other seasons" is quite speculative.

10) The trends on frequency of trajectories, by using only the geographical sector, where the starting point is 7 days before, it could give misleading results. However, potential trends in backtrajectories frequency constitutes a key point in the analysis and assessments of the paper.

11) EACOt (page 11 Line 9) does not seem to have any bearing on the changing trend of ozone, according to Figure 5 and 6.

12) In sections 3.2 and 3.3., it is difficult to understand why the authors have not used in-situ ancillary observations to distinguish the impact of direct ozone transport from that formed from precursors, and ozone from upper troposphere from pollution-derived ozone. Authors have used in a very limited way carbon monoxide (CO) in Section 3.3 (this does not appear in section 2.1 Data) but they have not crossed O3 and CO data to discriminate the O3 origin, but they have used the CO and backtrajectories trends (??). Authors might have also used water vapour mixing-ratio or absolute humidity to discriminate high ozone from upper levels. On the contrary, the authors have used rough simulations whose uncertainty is not known.

13) In section 3.3, again the methodological approach used in the backtrajectory sectors might result in wrong results since air masses pass over different ozone precursors sources along their paths. Considering the start-point (origin) of the trajectory is too simplistic.

14) In section 4.1 (Stratosphere-to-troposphere transport and jet characteristics) the methodology approach is also quite weak. The authors use model simulations, when they could also/instead in-situ water vapour mixing ratio at WLG to discriminate upper troposphere (rather than stratospheric air masses) with the help of PV at a near WLG level . Unfortunately, the example given for March 30, 2012 is also not good since the values of O3 and PV do not correspond to upper troposphere air masses (and even less to stratospheric air masses). The 7 PVU at 250 hPa does not justify the impact of upper tropospheric air masses to WLG.

15) In section 4.1 an important conceptual issue it is not clear at all. The authors, when referring to STE air masses, mean to a quite jet or to baroclinic cut-off lows (or deep lows) associated to the position of the jet? It is is difficult to conceive the direct impact and of a quite jet on surface ozone at WLG, and if it so, the authors should demonstrate

this important result.

16) Finally, the link between ozone at WLG with different modes of atmospheric circulation (section 4.2) is not justified or explained in all the cases. The authors limit themselves to presenting a series of statistical relationships, in some cases with very low and non-significant correlations, between ozone and climatic indexes, without necessarily having a causal relationship. Authors should decide whether to maintain this section with the degree of development they have so poorly achieved. If they maintain the section, it should be significantly improved, discarding those indices that clearly have no direct relation to the ozone observed in WLG.

Technical corrections It does not make sense to go into details without having deeply addressed the changes proposed in the major comments. English should be significantly smoothed as it is difficult to understand the meaning of some sentences of the manuscript.
* * *

---

## Referee Comment (RC2) · Anonymous Referee #2 · 19 Sep 2017

This manuscript presents a detailed analysis on the interannual variability and long-term trends of surface ozone at the Mt. Waliguan (WLG) station for the period of 1994-2013. A number of approaches including backward trajectory, chemical transport model simulations, tropospheric ozonesonde dataset, correlations with multiple climate modes, and multi-variable regression are applied to address this issue. The results identify the importance of stratosphere-troposphere exchange to the observed ozone increases at WLG in spring, and increasing influences of anthropogenic pollution from Southeast Asia in summer.

[Figure]

This study provides valuable information to better understand the long-term changes of surface ozone at a background station in western China. I also feel difficult to follow while reading the manuscript, and I understand the attempts to combine together all these different approaches and difficulty in assessing their inconsistency quantitatively. I have a few comments listed below for helping authors to clarify the manuscript.

**Specific comments**
1) Page 5, Line 5:
It is not clear how you clustered the trajectory directions into 45-degree bins. It shall be helpful to plot and define these bins on a figure, such as on a panel of Figure 1.
2) Page 8, Line 5:
For the statement "During summer, when air-masses from the east occur most frequently, the entire eastern sector reveals low PSCF", I suggest add "(as will be shown in Figure 2)" after "from the east occur most frequently", so that readers understand how you make the statement.
3) Page 8, Line 26:
Why do you state "the anthropogenic influence is negligible in all seasons except summer"? From Figure 1, we can also see high anthropogenic influences from Sichuan in spring and fall.
4) Page 11, Line 17-18:
You have argued above that the ozone trend in spring at WLG is driven by stratosphere-troposphere-exchange. If so, shall we expect filtering for the East Asian anthropogenic influences, i.e., air masses with lower stratospheric influences, would show a lower trend? However, the results here show nearly no change in the springtime trend. Can you explain?
5) Page 11, Line 20-30:
This section has showed that stratospheric influences explained two thirds of the ozone trend in spring. How about the rest one third? Would changes in anthropogenic emissions be the cause?

6) Page 12, Sect. 3.3:

The TOST dataset are monthly averages from 1994 to 2012. Does that mean the dataset already account for ozone changes associated with increases in East Asian anthropogenic emissions? And then the direct tropospheric ozone transport as calculated in this section (Figure 8 and 9) has considered the tropospheric ozone changes associated with increases in precursor emissions. Please clarify.

7) Page 13, Line 16-22:

In this paragraph, CO measurements at WLG are used to analyze the influences of anthropogenic emissions. The results show statistically significant increasing trends only in summer. How about the trend in autumn? The previous section showed that the ozone trend in autumn was driven by anthropogenic pollution, but this did not seem to be supported by the CO analysis. Can you please clarify? As for the contribution from precursor emissions, can the model simulation with fixed anthropogenci emissions provide a better estimate?

8) Page 13, Line 23:

I feel confused about the discussion on ozone trends based on different trajectories in this paragraph. It reported the largest ozone trend associated with the SE direction, and the lowest trend with the NW direction. However, back on Page 9, Line 25-30, the trajectory analysis showed that the NW trajectories associated with high ozone concentrations had increasing occurrence frequencies, while the SE trajectory frequencies were decreasing. Are they consistent?

9) Page 26, Table 5:

How do you estimate the ozone transported from East Asia, Europe, and North America? Please clarify.

**Some other comments**

1) Page 3, Line 30:

"GOES-Chem" should be "GEOS-Chem"?

2) Page 8, Line 25:

"and least so in summer". Need to remove "so"?

3) Page 9, Line 19:

Here alpha is used to denote statistical significance, while in a few other places, such as Page 10, Line 12, 'p' is used. Please make them consistent.

———————————————————

---

## Author Comment (AC1) · 31 Oct 2017

**Response to reviewer #2**

Thank you for your comments and suggestions. We have addressed all the issues that were raised and have made changes accordingly. We have also improved the English language and made other necessary changes. Please see our point-by-point response below.

**General comments**

This manuscript presents a detailed analysis on the interannual variability and long-term trends of surface ozone at the Mt. Waliguan (WLG) station for the period of 1994-2013. A number of approaches including backward trajectory, chemical transport model simulations, tropospheric ozonesonde dataset, correlations with multiple climate modes, and multi-variable regression are applied to address this issue. The results identify the importance of stratosphere-troposphere exchange to the observed ozone increases at WLG in spring, and increasing influences of anthropogenic pollution from Southeast Asia in summer.

This study provides valuable information to better understand the long-term changes of surface ozone at a background station in western China. I also feel difficult to follow while reading the manuscript, and I understand the attempts to combine together all these different approaches and difficulty in assessing their inconsistency quantitatively.

I have a few comments listed below for helping authors to clarify the manuscript.

**Specific comments**

1) Page 5, Line 5:

It is not clear how you clustered the trajectory directions into 45-degree bins. It shall be helpful to plot and define these bins on a figure, such as on a panel of Figure 1.

RE: Thank you for the suggestion. Instead of trajectory start-point direction we have calculated average trajectory directions, which have been used for clustering. To clarify the calculation process, we added the following example to show how the average 24h and 168h directions have been calculated as a supplement figure. The 45-degree bins were depicted along with the example.

We have made some changes in the second paragraph of section 2.2: "To study the overall air-mass origin and to determine whether the air-mass collected pollutants from the nearby cities, the average direction of each trajectory relative to the WLG station is calculated both for the 168h and for the 24h trajectory (Figure S1). The 168h and 24h average directions relative to WLG are clustered into bins of 45° and the occurrence frequency in each bin is calculated."

[Figure]

Figure 1 Schematic showing an example of the calculation process of 24h and 168h average trajectory directions and the 45-bins the trajectories were clustered into. The blue line shows a 7day trajectory example that bends from W to E, accounting for all the 168 hours, the average direction is westerly, while accounting only for the first 24 hours, the direction is easterly.

2) Page 8, Line 5:

For the statement "During summer, when air-masses from the east occur most frequently, the entire eastern sector reveals low PSCF", I suggest add "(as will be shown in Figure 2)" after "from the east occur most frequently", so that readers understand how you make the statement.

RE: Thank you for this suggestion. This sentence has been changed to "During summer, when air-masses from the east occur most frequently (as will be shown in Figure 2), the entire eastern sector reveals low values of high ozone PSCF, hardly showing signs of anthropogenic influence on WLG."

3) Page 8, Line 26:

Why do you state "the anthropogenic influence is negligible in all seasons except summer"? From Figure 1, we can also see high anthropogenic influences from Sichuan in spring and fall.

RE: Thank you for pointing out this. Page 8 lines 26-27 has been rephrased as:

"From the t=-168h average trajectory direction frequencies, it can be seen that the anthropogenic influence is strongest in summer, followed by autumn, and almost negligible in winter."

4) Page 11, Line 17-18:

You have argued above that the ozone trend in spring at WLG is driven by stratosphere troposphere-exchange. If so, shall we expect filtering for the East Asian anthropogenic influences, i.e., air masses with lower stratospheric influences, would show a lower trend? However, the results here show nearly no change in the springtime trend. Can you explain?

RE:We have explained this in the revised manuscript:

"We do not expect a decrease in springtime ozone when filtering the model for the East Asian anthropogenic influence, i.e., air masses with lower stratospheric influence, owing to the offsetting effects of increasing East Asian emissions."

5) Page 11, Line 20-30:

This section has showed that stratospheric influences explained two thirds of the ozone trend in spring. How about the rest one third? Would changes in anthropogenic emissions be the cause?

RE: Good suggestions. We have clarified in the revised manuscript:

"The stratospheric influence can explain two thirds of the total ozone increase at WLG in spring, with increases in Asian emissions contributing the rest one third"

6) Page 12, Sect. 3.3:

The TOST dataset are monthly averages from 1994 to 2012. Does that mean the dataset already account for ozone changes associated with increases in East Asian anthropogenic emissions? And then the direct tropospheric ozone transport as calculated in this section (Figure 8 and 9) has considered the tropospheric ozone changes associated with increases in precursor emissions. Please clarify.

RE: The TOST dataset is based on trajectory-mapped ozone soundings (Liu et al., 2013). The monthly averages of ozone in each grid should contain signals of background ozone and ozone produced with the grid from precursors emitted by anthropogenic and natural sources. Therefore, all the mean values in the TOST dataset already account for ozone changes associated with increases in East Asian (and other regions) anthropogenic emissions. One of the key issues in producing the TOST dataset was the impact of ozone production along the trajectories, which might cause errors in the mapped ozone data. A careful assessment indicates that the errors are mostly small and insignificant, as shown in Fig. 2 in Liu et al. (2013). Our approach of using TOST data is similar to the forward mapping in Liu et al. (2013), with the

difference that we focus on the impacts on ozone in the WLG grid from the surrounding grids. Therefore, it is likely that the impact of ozone production along the trajectories on our results is small, as in the case of Liu et al. (2013).

To clarify this we have added the following two paragraphs in section 3.3:

"Different from the GFDL-AM3 FIXEMIS simulation discussed in Section 3.2, the TOST approach discussed in this section does not eliminate the impacts from increases in Asian anthropogenic emissions. The TOST dataset is based on trajectory-mapped ozone soundings (Liu et al., 2013). The monthly averages of ozone in each grid should contain signals of background ozone and ozone produced within the grid from precursors emitted by anthropogenic and natural sources. Therefore, mean values in the TOST dataset account for not only ozone changes due to transport but also ozone changes associated with varying global-to-regional anthropogenic and natural emissions."

"One of the key issues in producing the TOST dataset was the impact of ozone production along the trajectories, which might cause errors in the mapped ozone data. A careful assessment indicates that the errors are mostly small and insignificant (Liu et al., 2013). Our approach of using TOST data is similar to the forward mapping in Liu et al. (2013). Therefore, it is likely that the impact of ozone production along the trajectories during their residence time on our results is small, as in the case of Liu et al. (2013).As the bottom layer of the grid in which WLG resides is excluded in our calculations, direct impacts on our results from regional emissions in the grid containing WLG can be ruled out."

Liu, G., Liu, J., Tarasick, D. W., Fioletov, V. E., Jin, J. J., Moeini, O., Liu, X., Sioris, C. E., and Osman, M.: A global tropospheric ozone climatology from trajectory-mapped ozone soundings, Atmospheric Chemistry and Physics, 13, 10659-10675, 10.5194/acp-13-10659-2013,2013.

7) Page 13, Line 16-22:

In this paragraph, CO measurements at WLG are used to analyze the influences of anthropogenic emissions. The results show statistically significant increasing trends only in summer. How about the trend in autumn? The previous section showed that the ozone trend in autumn was driven by anthropogenic pollution, but this did not seem to be supported by the CO analysis. Can you please clarify? As for the contribution from precursor emissions, can the model simulation with fixed anthropogenic emissions provide a better estimate?

RE: The CO data used in the manuscript are monthly CO data from weekly flask sampling (at 5m height) and analysis. A study by Zhang et al. (2011) indicates that the concentration of CO at WLG is subject to influences of regional-scale pollution, particularly in summer. Therefore, our summer CO measurements are less representative of large-scale conditions. In Section 3.2 we study the impacts from

anthropogenic emissions on a large-scale using GFDL-AM3 modeling. Our results are based on comparison of time-varying (BASE) and constant anthropogenic emissions (FIXEMIS).

8) Page 13, Line 23:

I feel confused about the discussion on ozone trends based on different trajectories in this paragraph. It reported the largest ozone trend associated with the SE direction, and the lowest trend with the NW direction. However, back on Page 9, Line 25-30,the trajectory analysis showed that the NW trajectories associated with high ozone concentrations had increasing occurrence frequencies, while the SE trajectory frequencies were decreasing. Are they consistent?

RE: These two results are not contradictory. The ozone concentrations associated with the NW trajectories are high in comparison to other sectors, but they show weak increasing trends, suggesting ozone coming from the NW have not changed much. The increase in NW trajectory frequency is what leads to the result that we experience these high ozone values more often, hence observe an increase in the ozone level. On the other side, ozone concentrations associated with the SE sector are not as high, but they show an increasing trend, highly possibly due to the change in precursor emissions in that sector.

9) Page 26, Table 5:

How do you estimate the ozone transported from East Asia, Europe, and North America? Please clarify.

RE: The GFDL model models CO-like-tracers from East Asia (EACOt), Europe (EUCOt) and North America (NACOt). For each month, we calculate the average ozone value associated with the upper 33 percentile EACOt, EUCOt, NACOt, to represent the ozone transported from East Asia ($O_{3,ea}$), Europe($O_{3,eu}$), and North America($O_{3,na}$). Then we use the East Asian Summer Monsoon Index (EASMI) to filter out those ($O_{3,ea}$,$O_{3,eu}$ and$O_{3,na}$) associated with the lower and upper 15 percentile of EASMI. The relative change induced by the East Asian Monsoon is then calculated with the equation: $(O_{3,\,EASMI} <= 15^{th} - O_{3,EASMI} >= 85^{th})/\bar{O}_3$.

We have clarified it in the revised manuscript by adding the above information to the header of Table 5.Thank you for pointing it out.

**Some other comments**

1) Page 3, Line 30:

"GOES-Chem" should be "GEOS-Chem"?

RE: Thank you, we have corrected this typo.

2) Page 8, Line 25:

"and least so in summer". Need to remove "so"?

RE: Thank you for the correction, we have made according change in the revised manuscript.

3) Page 9, Line 19:

Here alpha is used to denote statistical significance, while in a few other places, such as Page 10, Line 12, 'p' is used. Please make them consistent.

RE: Thank you for the suggestion, we have made it consistent throughout the manuscript.

We have tried to address the issues raised by both referees and improved the English language. We have also made other changes where necessary. The title of this paper has been changed to "Long-term trends of surface ozone and its influencing factors at the Mt. Waliguan GAW station, China – Part 2: The roles of anthropogenic emissions and climate variability".

---

## Author Comment (AC2) · 31 Oct 2017

**Response to reviewer #1**

Thank you for your comments and suggestions. We have addressed all the issues that were raised and have made changes accordingly. We have also improved the English language and made other necessary changes. Please see our point-by-point response below.

**General comments**

This paper explores the factors driving the observed ozone changes at Mt. Waliguan Observatory (WLG) using basically backward trajectory analysis and chemistry-climate model hindcast simulations (GFDL-AM3). The paper also deals links of ozone variability at WLG with the QBO, NAO, the East Asian summer monsoon (EASM), and the sunspot cycle. Although the paper addresses very interesting topics (probably too many issues in a single paper), complementary to that addressed in the companion paper (Xu et al., 2016), in a region of enormous interest such as the Tibetan Plateau, and using valuable data from a global GAW station such as WLG, the paper suffers from significant weaknesses that must be addressed withmore credible and robust approaches.

The most important drawbacks are of methodological nature, and are briefly discussed below.

**Specific comments:**

1) The approach used for the backtrajectories dataset and climatology does not seem the most advisable to distinguish between ozone long-range transport from ozone produced by regional precursors. The use of the directions of only start-points (origin) of the trajectories into bins of 45°is a very weak approximation. Air masses normally move among sectors along their entire trajectory (especially those of 7 days duration). So, it seems more reasonable to use some index accounting for the time of residence of the trajectory in each geographical sector. Bins of 45°seem to be too narrow for 7-day backtrajectories for which a great error / uncertainty in the geographical determination is associated.

RE: It is true that trajectories usually cross various sectors, which is why we analyzed both the 7day and 24h trajectory directions. The aim of using 7day direction is to look at the overall air-mass origin, while the 24h directions can show whether the air mass has changed its course before arriving at WLG. To better account for the other geographical sectors that the long trajectories might pass on their paths, instead of the direction of the t=-24h and t=-168h trajectory start-points, we now use the vector mean direction during the first 24h and 168h for each trajectory. The occurrence

frequencies in each direction bin were recalculated and results turned out to be quite similar to that in manuscript.

We also took the advice of the reviewer and made a comparison using the PBL and free tropospheric residence time to do the analysis in Figure 2 and Table 1 (see Figs.1-2). Results turned out to be similar to those based on start-points. The use of the residence time has its advantage and disadvantage. We not only want to know where the air mass has been, but also when the air mass has been there, which is why we use the direction of the trajectory track points.

[Figure]

Figure 1 The average trajectory direction occurrence frequencies in a) spring, b) summer, c) autumn and d) winter of 1) t<24h and 2) t<168h.

Figure 2 The 1) PBL and 2) free tropospheric trajectory residence time occurrence frequencies in a) spring, b) summer, c) autumn and d) winter.

We have redrawn Figures 2 and 3 using the statistics of average trajectory directions and updated the values in Tables 1 and 3 in the revised manuscript. We have added Figure S1 in the supplement to schematically show the way of obtaining average directions of the 168h and 24h trajectories. The 2nd paragraph in Section 2.2 has been modified as follows:

"To study the overall air-mass origin and to determine whether the air-mass collected pollutants from the nearby cities, the average direction of each trajectory relative to

the WLG station is calculated both for the 168h and for the 24h trajectory (Figure S1). The 168h and 24h average directions relative to WLG are clustered into bins of 45° and the occurrence frequency in each bin is calculated."

2) The use of 1-day trajectories to estimate the impact of regional ozone sources and those of 7-day path (very long) as representative of ozone long-range transport are not well understood and not sufficiently justified. In fact, when the 1994-2013 climatology of air mass origins at WLG in the PBL and FT are depicted (Figure 1), the main patterns in the distribution of the air masses frequency is quite similar for both regions (PBL and FT). Indeed, that means that the discrimination between PBL and FT air masses has not been satisfactorily achieved.

RE: As already pointed out in previous studies (Ma et al., 2002; Xu et al., 2016), during daytime and nighttime the WLG site is mainly influenced by air from the PBL and FT, respectively, causing a daytime minimum and a nighttime maximum of the ozone concentration. And ozone in daytime and nighttime showed different trends, particularly in summer when daytime and nighttime ozone showed respective trends of 0.07 ppb/yr (p=0.41) and 0.22 ppb/yr (p=0.04) (see Xu et al., 2016). To understand this difference, we think it is necessary to investigate the impacts of air masses from the PBL and FT separately. Observations at WLG, a high mountain site (3.8km asl) with very little local emissions represent the large-scale atmospheric conditions. Even the PBL air mass should mostly be representing the background condition. In addition, air masses in the PBL and FT often move in similar directions, particularly when they are driven by large-scale circulations. These explain why there are similarities between the PSCF of PBL and FT air masses. Nevertheless, there are some differences between the PSCF of PBL and FT air masses, for example, the high PSCF of FT air masses in the northeast sector (Figs. 1a2 and 1b2), the high PSCF of PBL air masses over Nepal and Northern India (Fig. 1d1), the much larger extension for the PSCF of FT air masses than that of PBL air masses, etc. Therefore, the separation of the PBL and FT air masses does provide some more details about the potential sources of ozone at WLG.

We have revised the last paragraph of Section 2.2 as " Since ozone is a trace gas with a distinct vertical distribution, it is not enough to just determine the direction from which the air-mass came. The height of the air-mass is also crucial for interpreting the measured ozone concentrations. As discussed in previous studies (Ma et al., 2002;Xu et al., 2016), the WLG site is predominantly influenced by air from the planetary boundary layer (PBL) during daytime and from the free troposphere (FT) during nighttime, with ozone concentrations showing a daytime minimum and a nighttime maximum. Daytime and nighttime ozone at WLG show different trends, particularly in summer (0.07±0.18 ppb year-1for daytime and 0.22±0.20 ppb year-1 for nighttime; Xu et al., 2016). To investigate the impacts of air masses from the PBL and FT separately, the PBL height, which can be added in the Hysplit model along the trajectories, is used to judge whether the air-mass that arrived at WLG is representing the PBL or the FT. PBL trajectory sections are defined as the part of the trajectory

that was continuously within the PBL before arriving at the station. Thus, PBL trajectory sections are usually close to the station. When the trajectory height exceeds that of the PBL, the rest of the trajectory is taken as the FT trajectory section. FT trajectory sections can also be close to the station, representing subsiding air from the FT near the station, however, most of them are located far away from the station."

Ma, J., Tang, J., Zhou, X., and Zhang, X.: Estimates of the Chemical Budget for Ozone at Waliguan Observatory, Journal of Atmospheric Chemistry, 41, 21-48, 10.1023/A:1013892308983, 2002.

Xu, W., Lin, W., Xu, X., Tang, J., Huang, J., Wu, H., and Zhang, X.: Long-term trends of surface ozone and its influencing factors at the Mt Waliguan GAW station, China – Part 1: Overall trends and characteristics, Atmospheric Chemistry and Physics, 16, 6191-6205, 10.5194/acp-16-6191-2016, 2016.

3) Nothing is said about the methodology used to determine the critical height of the back-trajectory in relation to the PBL height for each point of the airmass trajectory.

RE: The Hysplit model we used can add meteorology output (such as the PBL height, PBLH) along trajectories. The PBLH values along the trajectories are then compared with the trajectory height data, to determine whether the air mass is within the PBL or in the free troposphere. We have clarified this in Sect. 2.2 in the revised manuscript:

"To investigate the impacts of air masses from the PBL and FT separately, the PBL height, which can be added in the Hysplit model along the trajectories, is used to judge whether the air-mass that arrived at WLG is representing the PBL or the FT."

4) In page 8 Lines 5-13; The results are inconsistent and, in some cases, contradictory. Section 3.1 is plenty of inconsistencies such as the following in page 8 lines 16-18:"The t=-168h trajectory direction provides us information on the overall origin of the air-mass, while the trajectory direction calculated for t=-24h should be able to reveal if the airmass passed over nearby polluted regions before arriving at the station", while in lines 26-27, is said: "From the t=-168h trajectory direction frequencies, it can be seen that the anthropogenic influence is negligible in all seasons"

RE: Page 8 lines 5-13 has been modified as:

"During summer, when air-masses from the east occur most frequently(as will be shown in Figure 2), the entire eastern sector reveals low values of high ozone PSCF, hardly showing signs of anthropogenic influence on WLG. In other words, most air-masses from the east in summer are not associated with high ozone. High ozone PSCF occurs dominantly with trajectories from the NW or N. In autumn, in addition to NW or N, significant contributions of trajectories from the E, SE and S can also be discerned in the PBL trajectories, which suggest that high ozone is linked to air-masses coming from western China, central China and the northeastern part of the

Tibetan Plateau, the southwestern part of Gansu province as well as north of China (east Mongolia). In the FT trajectories, high ozone concentrations were mainly linked to air-masses from western and central China. In addition, air masses over Gansu province, part of the Sichuan province and some parts of Russia also show high PSCF. In winter, the PBL trajectories show high ozone PSCF mainly in the NW sectors, however, the SW and N-NE sectors also revealed scattered high PSCF values (over some parts of Nepal, Northern India, Mongolia and Inner Mongolia). Aside from the NW sector, the FT trajectories display significantly high PSCF in the NE sector in the western half of Inner Mongolia."

Page 8 lines 26-27 has been rephrased as:

"From the 168h average trajectory direction frequencies, it can be seen that the anthropogenic influence is strongest in summer, followed by autumn, and almost negligible in winter."

5) Analyzing 24h and 7-day trajectories, how it is possible to say that "....with PBL airmasses dominating during the day and FT air-masses during the night, which led to a clear diurnal variation of high nighttime and low daytime ozone concentrations". This situation, which is very realistic, probably overturns all the assumptions made for the establishment of the methodology of FT and PBL backtrajectories.

RE: The conclusion "....with PBL air-masses dominating during the day and FT air-masses during the night, which led to a clear diurnal variation of high nighttime and low daytime ozone concentrations" was not drawn from the analysis of 24h and 7-day trajectories, it is the finding in studies by Ma et al. (2002) and Xu et al. (2016).

This conclusion is not in contradiction with the methodology used for the backtrajectories. Free-tropospheric air masses over WLG during nighttime might have come from the PBL of upwind locations, whereas PBL air masses during daytime might have also come from the free troposphere. The time of the day cannot provide enough information on the overall characteristics of the air mass, which is why we seek the aid of backtrajectories.

6) All of Section 3.1 should be reviewed using a consistent methodology.

RE: We thank the reviewer for the suggestion. We have carefully gone through and revised this section.

7) In Section 3.2 is difficult to support a joint analysis of point observations in WLG with simulations of GFDL-AM3 with a resolution of 200X200 km2.

RE: We respectfully disagree with the reviewer for this statement. Observations at the 3.8km altitude of WLG in the remote atmosphere of the Tibetan Plateau are representative of large-scale conditions that a 200x200 $km^2$ global model is expected to resolve. It is appropriate to compare observations at WLG with the model

simulations sampled at 700 hPa. We have clarified this in Section 2.5 in the revised manuscript:

"The long-term ozone observational record at WLG provides an important test for the GFDL-AM3 model to represent the key processes driving year-to-year variability and trends of tropospheric ozone in the remote atmosphere of the Tibetan Plateau. For comparison with measurements at the 3.8 km altitude of WLG, the model is sampled at the grid box containing WLG and at the 700 hPa layer. This approach is appropriate because observations at Mt. WLG are representative of large-scale conditions with little influence from local urban emissions."

8) I do not see GFDL-AM3 captures the inter-annual variation of observed surface ozone anomaly, with the correlation coefficient ranging from 0.5 to 0.7 for spring, summer and autumn, as it is said.

RE: We have rephrased the discussion in the revised manuscript:

"GFDL-AM3 captures some inter-annual variation of observed surface ozone anomaly, with the correlation coefficient ranging from 0.5 to 0.7 for spring, summer and autumn. The correlations between the observed and modelled ozone anomaly are significant at the 90% confidence level in all seasons except winter. The model fails to reproduce the small observed ozone variability in winter."

9) The sentence "A stratospheric ozone tracer implemented in GFDL-AM3 (O3Strat; Sect. 2.5) indicates that the stratospheric influence can explain 23% (r=0.48) of the observed ozone interannual variability in spring (Fig.4a) but contributes little to observed variability in other seasons" is quite speculative.

RE: We have clarified the credibility of AM3 O3Strat to infer stratospheric influence, based on prior process-oriented evaluation with intensive field measurements available over the western United States:

"A stratospheric ozone tracer implemented in GFDL-AM3 ($O_3$Strat; Sect. 2.5) enables us to quantify the stratospheric contribution to variability and trends of ozone measured at WLG. Prior analysis of daily ozonesondes, water vapour, and lidar measurements indicates that variability in AM3 $O_3$Strat represents the episodic, layered structure of ozone enhancements in the free troposphere consistent with the observed characteristics of deep stratospheric intrusions (Lin et al., 2012b; Lin et al., 2015a; Langford et al., 2015). Sampling AM3 $O_3$Strat at WLG indicates that the stratospheric influence can explain 23% (r=0.48) of the observed ozone interannual variability at WLG in spring (Fig.4a) but contributes little to observed variability in other seasons (r<0.1; Fig.4b-d)."

This conclusion is further supported by our model sensitivity simulations with time-varying and constant anthropogenic emissions (Fig.6 and related discussions in the text).

10) The trends on frequency of trajectories, by using only the geographical sector, where the starting point is 7 days before, it could give misleading results. However, potential trends in backtrajectories frequency constitutes a key point in the analysis and assessments of the paper.

RE: As mentioned in the response to comment 1, the trajectory directions of t=-168h have been replaced by the average directions during the traevelling, which should be more representative of the geographical sectors on the pathways of the trajectories. The trajectory frequency trends were recalculated and results turned out to be similar with those in the previous manuscript.

11) EACOt (page 11 Line 9) does not seem to have any bearing on the changing trend of ozone, according to Figure 5 and 6.

RE: We are not sure what the reviewer means. Emissions of EACOt do not change over time. We are using these COt tracers to bin modelled ozone according to the dominant influence of different continental air regimes in the BASE simulation, in which emissions of ozone precursors change over time. That sentence is rephrased:

"To evaluate the effect of pollution transport from Southeast and East Asia, we filter ozone in the AM3 BASE simulation with the East Asian CO tracer (EACOt; see Sect.2.5)."

12) In sections 3.2 and 3.3., it is difficult to understand why the authors have not used in-situ ancillary observations to distinguish the impact of direct ozone transport from that formed from precursors, and ozone from upper troposphere from pollution-derived ozone. Authors have used in a very limited way carbon monoxide (CO) in Section 3.3 (this does not appear in section 2.1 Data) but they have not crossed O3 and CO data to discriminate the O3 origin, but they have used the CO and backtrajectories trends (??). Authors might have also used water vapour mixing-ratio or absolute humidity to discriminate high ozone from upper levels. On the contrary, the authors have used rough simulations whose uncertainty is not known.

RE: The CO data used in the manuscript are monthly CO data from flask sampling and analysis. The time resolution of the data has limited its use. High resolution CO data would be much better to distinguish ozone measurements impacted by anthropogenic emissions and from those impacted by upper tropospheric/lower stratospheric air. Indeed, based on shorter period observations, Wang et al. (2006) were able to identify ozone measurements impacted by upper tropospheric/lower stratospheric air using the negative correlation between CO and ozone. In-situ observations of CO have been attempted at WLG using different techniques. Unfortunately, the coverage of qualified data is poor due to various technical problems. Therefore, we cannot use in-situ CO measurements in the analysis of the long-term ozone measurements (1994-2013). However, some reliable in-situ measurements of CO are available for recent years. We used these CO data to check the reliability of the model and found a significant correlation (r=0.48, p<0.01)

between the measured CO and the modeled East Asian COtracer, which proves that the model is able to identify pollution transport from East Asia.

We tried using water vapor data to identify air mass from upper layers. However, the RH data from the WLG station show significant, uncorrectable biases during the period of 2004-2013, which influence the credibility of the analysis outcome. Hence, this part of the study was not brought into the manuscript.

We have added in Section 2.1 "We also use monthly CO data based on weekly flask sampling at 5 m above ground, obtained from the World Data Centre for Greenhouse Gases (http://ds.data.jma.go.jp/gmd/wdcgg/wdcgg.html), to infer changes of regional emissions near WLG. Using high frequency (e.g., minutes) in-situ observations of CO and water vapour would be ideal to diagnose the presence of stratospheric versus anthropogenic influences, however, continuous high-quality data are not available at WLG due to various technical challenges." in the revised manuscript.

Wang, T., Wong, H. L. A., Tang, J., Ding, A., Wu, W. S., and Zhang, X. C.: On the origin of surface ozone and reactive nitrogen observed at a remote mountain site in the northeastern Qinghai-Tibetan Plateau, western China, Journal of Geophysical Research: Atmospheres, 111,25 D08303, 10.1029/2005JD006527, 2006.

13) In section 3.3, again the methodological approach used in the backtrajectory sectors might result in wrong results since air masses pass over different ozone precursors sources along their paths. Considering the start-point (origin) of the trajectory is too simplistic.

RE: Indeed, if we want to consider all the sectors on the trajectory pathway, the use of trajectory directions has its limitations. However, we do not know the actual 4D distribution of ozone, hence considering all the sectors of the trajectory has no sense. Based solely on the ozone observations at WLG, using the backtrajectories to filter out air masses that might be influenced by anthropogenic emissions is the most direct method. As an improvement to the methodology, we already replaced the trajectory directions calculated from the start-point with average trajectory directions, which should be more representative of all the sectors on the trajectories. Results turned out to be quite similar with the previous ones.

14) In section 4.1 (Stratosphere-to-troposphere transport and jet characteristics) the methodology approach is also quite weak. The authors use model simulations, when they could also/instead in-situ water vapour mixing ratio at WLG to discriminate upper troposphere (rather than stratospheric air masses) with the help of PV at a near WLG level . Unfortunately, the example given for March 30, 2012 is also not good since the values of O3 and PV do not correspond to upper troposphere air masses (and even less to stratospheric air masses). The 7 PVU at 250 hPa does not justify the impact of upper tropospheric air masses to WLG.

RE: We tried using water vapor data to identify air mass from upper layers, however,

the RH data at the WLG station show significant errors during the period of 2004-2013, which influence the credibility of the analysis outcome. Hence, this part of the study was not brought into the manuscript. We clarify this in Sect. 4.1 of the revised manuscript:"Due to the transient, localized nature of stratospheric intrusions, diagnosing the presence of stratospheric influence in near-surface ozone requires precise, high-frequency (a few minutes), and co-located measurements of ozone, CO, water vapor and surface wind gust at remote sites (see Langford et al., 2015). These measurements are not available at WLG. Thus, we rely on a global model that has been previously shown to be able to represent deep stratospheric intrusions".

In the example given for March 30, 2012, we did not solely look at the PV at 250hPa over WLG to prove that it is an STT event. We made slight changes to Figure 12 to clarify the case (see Figure 3). 500hPa Geopotential height, the U wind cross-section were added and we have changed the color scales in Figure 12c-d to more clearly show this event of stratospheric intrusion. The isentropic PV filament is clearly depicted in Figure 3b. The high PV airmass was transported south to the midlatitudes by the strong northerly winds ahead of the ridge and behind the trough. The easterly airflow between 450hPa and 250hPa north of the fold and the subtropical jet south of the fold have brought the high PV airmass to the west to WLG (Figure 3c). Figure 3c clearly showing a tropopause fold over WLG, The high PV reached down to the surface layer at WLG station and correspondingly the transport of stratospheric high $O_3$ down to the surface can be detected in Figure 3d.

[Figure]

Figure 3 a) Map of 500hPa geopotential height (white contours), 700hPa temperature (shading) and wind field (black arrows); b) Map of 250hPa potential vorticity; c) the cross-section of potential vorticity along the 101.0E longitude line. The white line denotes the 1 PVU isoline, the black lines are

U wind isolines (dashed lines for westerly winds and solid lines for easterly winds) and the red dots indicate the location of the subtropical jet stream (U wind>35m s-1); d) The cross-section of ozone mixing ratios, V wind and W wind vector along the 101.0E longitude line from the ECMWF reanalysis during an STT  transport event on 30 Mar 2012. The while line denotes the 50-ppbv ozone contour.

Langford, A. O., Senff, C. J., Alvarez Ii, R. J., Brioude, J., Cooper, O. R., Holloway, J. S., Lin, M. Y., Marchbanks, R. D., Pierce, R. B., Sandberg, S. P., Weickmann, A. M., and Williams, E. J.: An overview of the 2013 Las Vegas Ozone Study (LVOS): Impact of stratospheric intrusions and long-range transport on surface air quality, Atmospheric Environment, 109, 305-322, 2015.

15) In section 4.1 an important conceptual issue it is not clear at all. The authors, when referring to STE air masses, mean to a quite jet or to baroclinic cut-off lows (or deep lows) associated to the position of the jet? It is difficult to conceive the direct impact and of a quite jet on surface ozone at WLG, and if it so, the authors should demonstrate this important result.

RE: In the case study of 30[th] Mar 2012, the STE was associated with a deep low, which, however, is not always the case. Typical baroclinic cut-off lows were hardly observed during the STT events in the springs of 1999 and 2012. The frequency of stratospheric intrusions has been found to be highest along the subtropical jet stream, where the tropopause break is located (e.g. Homeyer, 2012 and Sprenger et al., 2003). Tropopause folds are typically located to the north of the subtropical jetstream, hence the location of the jet stream directly influences the location of the STE event. If the jet is located more to the south, the stratospheric ozone input might not reach WLG.

For the springs of 1999 and 2012, we calculated the average subtropical jet location (latitude) at the longitude of WLG for STE cases and non-STE cases (Table 1). Results show that the shift of the jet to the north pushes the location of the tropopause fold to the north, which then leads to STE processes over the WLG region. The difference in STE and Non-STE jet location passed the t-test at a 99% significance level.

Table 1 Average subtropical jet location (latitude) for STE cases and non-STE cases for the springs of 1999 and 2012

|  | Spring 1999 | | Spring 2012 | |
| --- | --- | --- | --- | --- |
|  | avg | std | avg | std |
| STE | 35.6 | 4.5 | 35.2 | 5.3 |
| NO-STE | 33.1 | 4.3 | 31.8 | 4.9 |

Homeyer, C. R.: Chemical and Dynamical Characteristics of Stratosphere-Troposphere Exchange, Atmospheric Sciences, Texas A&M University, 2012.

Sprenger, M. and Wernli, H.: A northern hemispheric climatology of cross-tropopause exchange for the ERA15 time period (1979-1993), J. Geophys. Res., vol. 108, no. D12, 8521, 2003.

16) Finally, the link between ozone at WLG with different modes of atmospheric circulation (section 4.2) is not justified or explained in all the cases. The authors limit themselves to presenting a series of statistical relationships, in some cases with very low and non-significant correlations, between ozone and climatic indexes, without necessarily having a causal relationship. Authors should decide whether to maintain this section with the degree of development they have so poorly achieved. If they maintain the section, it should be significantly improved, discarding those indices that clearly have no direct relation to the ozone observed in WLG.

RE: This paper is the part-2 of our study about long-term measurements of surface ozone at WLG. In the part-1 paper (Xu et al., 2016) ozone trends were obtained and the time-series of surface ozone at WLG was decomposed into five intrinsic mode functions (IMFs) with different periodicities. The IMFs did not contribute much to ozone trends but they are important in the interannual as well as seasonal variabilities. This paper aims to understand the factors driving the long-term trends and interannual variability. The major part of this paper focuses on the interpretation of the observed ozone trends. However, we think it is also necessary to understand the possible causes of the interannual variability. We tried this in the previous manuscript, but we did not show causal relationships. After careful consideration we decide to maintain this section but make necessary changes.

We have removed the materials about the NAO (Figure 15 and last paragraph in section 4.2 in previous manuscript) from this section. We have strengthened the analysis about the QBO and added more discussions in this section. We found that the QBO index was positively correlated with zonal and meridional wind over the areas west and north of China, suggesting increases in westerly and southerly winds over those areas when the QBO was in its positive phase. We also found similar positive correlations between the QBO index and air temperatures at different pressure levels, with a warming of 0.01–0.05 ℃ per unit increase in the QBO index. We think that these periodic changes in air circulations and temperature might have influenced the transport of ozone and its precursors and the photochemical conditions. Furthermore, we can see significant positive correlations of the QBO index with the 3-8 km TOST ozone columns over some areas west and north of China, over which the FT air can be transported to WLG. Therefore, we believe that the QBO can exert a small indirect influence on surface ozone at WLG through periodically changing dynamical and photochemical conditions over west and north of China.

The first paragraph of this section has been revised as follows:

[revised manuscript text omitted]

Figure 14 Correlation coefficients between the QBO index and zonal (a, c) and meridional (b, d) wind at 500 hPa and 700 hPa with grey dots indicating those that are significant (p<0.05). The red triangles indicate the position of WLG.

[Figure]

Figure 15 Correlation coefficients between the QBO index and the 3-8 km TOST ozone columns. Correlations for the grids with grey dots are significant (p<0.05)

[Figure]

Figure S2 Regression slopes for the correlations between the QBO index and air temperatures at 200 hPa (a), 500 hPa (b), 700 hPa (c) and surface (d), with grey dots indicating the that are significantly correlated (p<0.05). The red triangles indicate the position of WLG.

**Technical corrections**

It does not make sense to go into details without having deeply addressed the changes proposed in the major comments. English should be significantly smoothed as it is difficult to understand the meaning of some sentences of the manuscript.

RE: We have tried to address the issues raised by both referees and improved the English language.